# IGF2BPs directly regulate the noncanonical translation of toxic proteins from mutant *FMR1* mRNA containing expanded CGG repeats

Anna Baud [1,3] ✉, Damini Saha [2], Tomasz Skrzypczak[2], Izabela Broniarek [1], Daria Niewiadomska [1], Wojciech J. Szlachcic [1], Malgorzata Borowiak[1], Rajani Kanth Gudipati[2] & Krzysztof Sobczak [1,3] ✉

Mutant mRNA of the fragile X messenger ribonucleoprotein 1 gene (*FMR1*) containing expanded CGG repeats in its 5'UTR is a primary cause of fragile X premutation associated conditions. It serves as a template for the biosynthesis of the major open reading frame encoding canonical protein and the downstream open reading frame containing expanded CGG repeats encoding toxic FMRpolyG protein that comprise a long polyglycine stretch, produced via repeat-associated non-AUG initiated translation. Here, we show that insulin-like growth factor 2 mRNA-binding protein 3 (IGF2BP3) binds directly to the 5'UTR of *FMR1* RNA, and the sequence in the vicinity of near-cognate start codons of non-AUG translation is pivotal for IGF2BP3 binding. Upon IGF2BP3's knockdown, FMRpolyG biosynthesis and cell toxicity evoked by FMRpolyG, significantly decreased in cells expressing mutant *FMR1* with expanded CGG repeats. Disruption of IGF2BP ortholog in novel fragile X premutation associated conditions *C. elegans* model rescues the disease phenotype induced by expression of a human *FMR1* RNA fragment containing expanded CGG repeats. Our results suggest that IGF2BP3 positively regulates the noncanonical translation of expanded CGG repeats and may be a promising target for clinical applications.

Short tandem repeats occur frequently within exons and noncoding regions of the human genome. They are polymorphic in length and, when long enough, prone to expansion. Expansions over a certain threshold within different regions of protein-coding genes have been linked to over 50 human-inherited repeat expansion disorders, mainly neurodegenerative or neuromuscular. One such genetic loci is the fragile X messenger ribonucleoprotein 1 gene (*FMR1*), in which CGG repeats occur in the 5'UTR. In healthy individuals, this locus contains between 5 and 54 CGGs. However, the expansion of CGG repeats (CGGexp) to intermediate size, called premutations (PMs, 55–200 CGG repeats), can cause different fragile X-PM-associated conditions (FXPACs)[1], including fragile X-associated tremor/ataxia syndrome (FXTAS), fragile X-associated primary ovarian insufficiency (FXPOI), and fragile X-associated neuropsychiatric disorders (FXANDs). Longer expansions, called full mutations (above 200 CGG repeats) lead to the development of early-onset neurodevelopmental fragile X syndrome

[1]Department of Gene Expression, Institute of Molecular Biology and Biotechnology, Adam Mickiewicz University, Poznan, Poland. [2]Center for Advanced Technologies, Adam Mickiewicz University, Poznan, Poland. [3]These authors jointly supervised this work: Anna Baud, Krzysztof Sobczak. ✉e-mail: anna.baud@amu.edu.pl; ksobczak@amu.edu.pl

(FXS). At the molecular level, a full mutation causes hypermethylation and silencing of the *FMR1* promoter, which results in the loss of the protein product, fragile X messenger ribonucleoprotein 1 (FMRP)[2].

FXTAS is a late-onset, incurable neurodegenerative disorder characterized by tremors, ataxia, cognitive decline, and Parkinsonian features[3]. PMs occur at a frequency of 1 in 150–300 females and 1 in 400–850 males in the general population, but due to the location of the *FMR1* gene on the X chromosome, male carriers are primarily affected. The prevalence of FXTAS is 40–75% of males with PM, and 16–20% of female carriers[4]. The pathological hallmark of FXTAS is the presence of large, ubiquitin-positive inclusions in the nuclei of neurons and astrocytes[4]. FXPOI symptoms include reduced fertility, premature menopause, and ovarian dysfunction[5], with a prevalence of 20–30% in females with PM[6]. FXAND manifests as anxiety and depression in adults[7]. The pathogenesis of FXPAC remains unclear, and several nonexclusive pathogenic mechanisms have been proposed. One involves the cotranscriptional formation of aberrant R-loops in the CGGexp locus triggering a DNA-damage response[8–10]. Another suggests a toxic RNA gain-of-function leading to the sequestration of proteins on RNA-containing CGGexp (rCGGexp) within nuclear foci[11–15]. This results in the functional depletion of these proteins and impairment of their physiological functions. The third mechanism underlying rCGGexp toxicity in FXPAC is the repeat-associated non-AUG initiated (RAN) translation of rCGGexp into aberrant proteins that contain stretches of repeated amino acids despite CGG repeats being located in the 5′UTR[16,17]. These three mechanisms can cause downstream effects, such as mitochondrial dysfunction, oxidative stress, impaired miRNA biogenesis, alternative splicing, or the presence of intranuclear inclusions, which contribute to the pathogenesis of FXPAC.

When *FMR1* mRNA containing short or long CGG repeats is exported to the cytoplasm, its translation is initiated at the AUG start codon downstream of the CGGs, resulting in the production of the main protein product, FMRP. However, rCGGexp embedded within the *FMR1* 5′UTR can also be translated via RAN translation, initiating at non-canonical, near-cognate start codons, resulting in the biosynthesis of toxic RAN proteins[16,17]. These proteins contain repeated glycine, alanine, or arginine tracts (FMRpolyG, FMRpolyA, and FMRpolyR, respectively), depending on the reading frame, or a combination of them as a result of translational frameshifting[18]. The FMRpolyG translation is the most efficient and initiates at the near-cognate ACG or GUG codons located upstream of the rCGGexp[16,17], while translation initiation of FMRpolyA probably occurs within the repeats (GCG codon)[16,19]. FMRpolyG is neurotoxic and tends to aggregate in cytoplasmic or ubiquitin-positive nuclear inclusions[20,21]. Its important role in FXTAS pathogenesis is underlined by the fact that FMRpolyG-positive aggregates have been identified in various tissues, including the frontal cortex, hippocampus, heart, kidney, adrenal gland, and thyroid of FXTAS patients[17,22,23], and the ovarian stromal cells of FXPOI patients[24]. Although *FMR1* expression is not limited to neurons, the toxicity of intranuclear inclusions appears to be most prominent in the central nervous system.

A number of studies have contributed to the understanding of the CGG RAN translation mechanism, but knowledge about this process remains limited. Recent reports have shown that the RAN translation of rCGGexp requires a 5′ cap and the translation initiation factors eIF4A and eIF4E for initiation[25]. Additionally, a hairpin formed by rCGGexp or a nearby 5′UTR sequence causes steric blockage for the scanning of the 43S preinitiation complex (PIC)[25,26]. This leads to an increased dwell time of 43S PIC, lowered codon fidelity, and initiation at near-cognate codons (ACG or GUG)[27]. RNA molecules containing expanded repeats and RAN proteins were shown to activate protein kinase R, which phosphorylates eIF2α and causes the activation of the integrated stress response[28]. This upregulates the RAN translation of rCGGexp and leads to a global translation shutdown. Genetic and proteomic studies have revealed that the efficiency of RAN translation may be regulated by

various proteins, including the translation initiation factors: eIF1 and eIF5; helicases: DEAD box helicase eIF4A[25], DDX3×[29], and DHX36[30]; small ribosomal subunit proteins: e25 and e26 (RPS25 and RPS26, respectively); and some of the ribosomal quality control factors: nuclear export mediated facilitator, E3 ligase listerin, ankyrin repeat, and zinc finger peptidyl tRNA hydrolase 1[31].

From a therapeutic perspective, identifying modifiers of RAN translation is likely to be crucial for the development of FXTAS treatment strategies. In this study, we employed a pull-down approach targeting rCGGexp and combined it with a proteomic analysis to identify proteins binding to the 5′UTR of *FMR1* mRNA in vitro. Selected candidates were further stratified to assess their contribution to RAN translation efficiency in FXTAS cell models. Among the selected proteins, we focused on insulin growth factor 2 mRNA binding protein 3 (IGF2BP3, also called IMP3), a nucleocytoplasmic protein composed of six RNA-binding domains: two N-terminal RNA-recognition motifs and four hnRNP K-homology domains connected by short linkers[32]. Studies performed using the *Xenopus laevis* model suggest that this protein plays a role in neuronal development[33]. Interestingly, IGF2BP3 has also been detected in human placentas and testes, indicating its potential role in reproductive organs. Depending on the target mRNA, IGF2BP3 can act as a modulator of mRNA cellular localization, RNA stability, miRNA biogenesis, or as an m6A reader[34]. Importantly, binding of IGF2BP3 to the 5′UTR of insulin growth factor 2 (*IGF2*) leader-3 mRNA activates its translation[35]. In this study, we show that IGF2BP3 directly binds to the 5′UTR of mutant *FMR1* mRNA and specifically promotes the RAN translation of toxic FMRpolyG. We also found that two other paralogs of the IGF2BP family, IGF2BP1 and IGF2BP2, can regulate FMRpolyG levels similarly to IGF2BP3, and their action is restricted to RAN-translated proteins. We propose that the inhibition of IGF2BP paralogs is a promising therapeutic approach to decrease the level of toxic RAN-translated FMRpolyG proteins in FXPAC. Finally, we show that IGF2BP3 knock down mitigates necrosis of FXTAS iPSC-derived neurons, and functional knock out of imph-1, an IGF2BP ortholog, improves the disease phenotype observed in the FXPAC *C. elegans* model.

## Results

### RNA-protein pull-down identifies sets of putative interactors of the 5′UTR of *FMR1* mRNA

To identify proteins that may bind to the 5′UTR of mutant *FMR1* mRNA, we performed a pull-down assay using a set of biotinylated RNAs, followed by a proteomic analysis. The bait RNAs included the 5′UTR of *FMR1* containing ca. 100 CGG repeats (FMR1-99CGG), the 5′UTR of *FMR1* devoid of CGG repeats (FMR1-delCGGs), control RNA of similar length and GC content (GC-rich), and RNA containing 23CGG repeats, without *FMR1* flanking sequence (23CGG) (Fig. 1A). First, we incubated in vitro transcribed, biotinylated RNAs with the protein extract of human neuroblastoma SH-SY5Y cells to assemble RNA-protein complexes, which were further purified on streptavidin-coated beads. Different elution profiles were observed on SDS-PAGE gels for different RNA baits (Supplementary Fig. 1A). Using mass spectrometry (MS), we identified ca. 25 proteins significantly enriched on FMR1-99CGG RNA compared to control GC-rich RNA (Fig. 1B, Supplementary Data 1). These proteins included Pur Beta, nucleophosmin, and heterogeneous nuclear ribonucleoprotein M (hnRNP M), which were previously described as binding to rCGGexp[11,13]. We also identified RPS26, which was previously shown to regulate the translation of FMRpolyG from the 5′UTR of mutant *FMR1* containing ca. 100 CGG repeats[36]. A gene ontology (GO) analysis of proteins binding to FMR1-99CGG RNA revealed significant enrichment of the following GO terms: structural constituent of the ribosome, translation, and mRNA-5′UTR binding (Supplementary Fig. 1B).

We then compared proteins binding to either FMR1-99CGG RNA or FMR1-delCGGs devoid of CGG repeats. A label-free analysis revealed

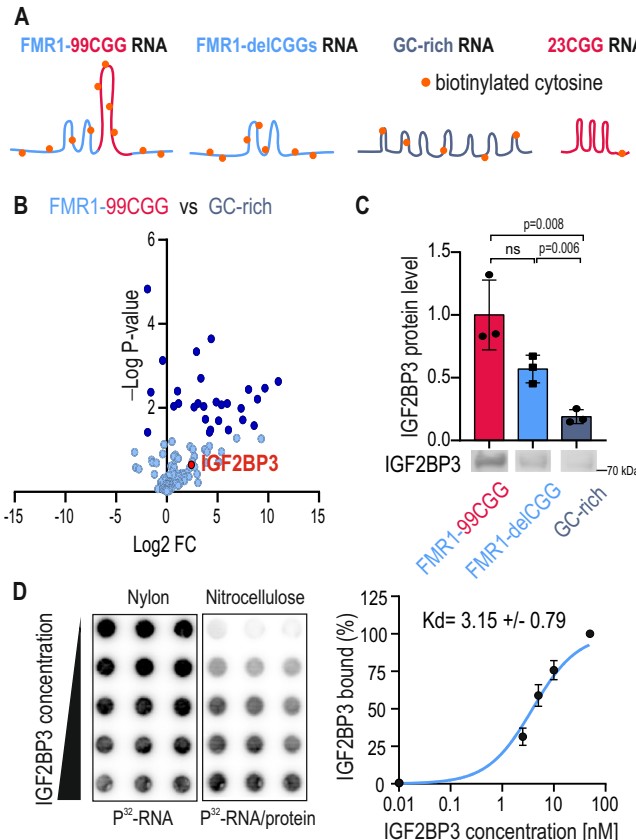

**Fig. 1 | Identification of proteins binding to FMR1-5'UTR RNA containing expanded CGG repeats. A** Schematic of in vitro transcribed biotinylated RNA molecules used in the study. The FMR1-99CGG RNA encodes the 5'UTR of *FMR1* (blue trait) with expanded CGG repeats forming a hairpin structure (magenta). The FMR1-delCGG RNA corresponds to the 5'UTR of *FMR1*, devoid of CGG repeats. The GC-rich RNA encodes *TMEM107* mRNA, characterized by high GC content (>70%; similar to *FMR1* RNA). The 23CGG corresponds to synthetic RNA probe composed of 23CGG repeats biotinylated at the 3' end. Orange dots represent randomly biotinylated cytosine residues. **B** Enrichment of proteins interacting with FMR1-99CGG RNA compared to GC-rich RNA and measured using label-free quantitative proteomics. Proteomes were compared using t-test statistics with a permutation-based FDR of 5%. Log2 fold-changes and *P*-values are given, indicating the magnitude of enrichment and statistical significance. The IGF2BP3 protein (red dot) is enriched on FMR1-99CGG RNA. **C** Quantification by Western blot of IGF2BP3 pulled down with three in vitro transcribed biotinylated RNA molecules. An unpaired two-sided t-test was used to calculate statistical significance: ns, non-significant. Note that in control experiment without any RNA in pull down no IGF2BP3 signal was detected (Supplementary Fig. 1). **D** The affinity of recombinant IGF2BP3 to radiolabeled FMR1-5'UTR RNA containing 16CGG repeats based on filter binding assay. **C**, **D** Mean values of the $N = 3$ independent experiments ± standard deviations (SD) are shown on the graph.

that more than half of the identified proteins were eluted with both RNAs (Supplementary Fig. 1C), including IGF2BP3. Twelve proteins were enriched more than two-fold on FMR1-delCGGs RNA, including multiple ribosomal proteins, and the most enriched protein being ubiquilin 2, which was previously detected in nuclear inclusions of FXTAS neurons[20]. Fifteen proteins preferentially bound to FMR1-99CGG, including nucleophosmin and nucleolin. Finally, we analyzed proteins binding to FMR1-99CGG RNA in comparison to 23CGG RNA probe missing *FMR1* flanking sequences (Supplementary Fig. 1D). In this dataset, we quantified 135 proteins binding to both RNAs, of those 50 proteins were significantly enriched on FMR1-99CGG RNA, including Pur Alpha, Pur Beta, HNRNP M, and IGF2BP3.

Among the identified proteins, IGF2BP3 protein, involved in RNA transport, translation, stability, and reading m6A-modified RNAs[34], drew our particular attention.

To validate the direct in vitro interaction of IGF2BP3 with FMR1-99CGG RNA and to decipher whether this interaction depends on CGG repeats, we used several orthogonal approaches. First, an RNA-protein pull-down followed by Western blot (WB) indicated that IGF2BP3 binds to both FMR1-99CGG RNA and FMR1-delCGGs mRNA, and, to a lesser extent, the control GC-rich RNA (Fig. 1C). Using filter binding and electrophoretic mobility-shift assays (Fig. 1D and Supplementary Fig. 1F), we confirmed that recombinant IGF2BP3 binds with high affinity to $^{32}$P-labeled FMR1-5'UTR RNA containing 16CGG repeats, with a low nanomolar dissociation constant (Kd = 3.15 ± 0.79 nM). We did not observe an effective interaction between IGF2BP3 and control $(UAA)_{20}$ RNA, even in the highest protein concentration range (Kd »100 nM) (Supplementary Fig. 1F). Finally, we analyzed publicly available enhanced UV-cross-linking and immunoprecipitation (eCLIP) data[37] and identified IGF2BP3 reads along FMR1 5'UTR (Supplementary Fig. 1G) in HepG2 cells containing normal size of CGG repeats, confirming our in vitro data. Together, our experiments confirmed that, under in vitro conditions, IGF2BP3 directly interacts with the FMR1 5'UTR in a CGG-independent manner.

## IGF2BP3 positively regulates the level of RAN-translated FMRpolyG in cells with expanded CGG repeats

To check whether IGF2BP3 regulates the RAN translation of FMRpolyG we first knocked down (KD) IGF2BP3 levels using short interfering RNA (siRNA) in human embryonic kidney cells (HEK-293T), human cervical carcinoma (HeLa), and human neuroblastoma cells (SH-SY5Y) with a transient expression of RAN-translated FMRpolyG containing tract of either 16 or 99 glycine residues tagged with green fluorescent protein, GFP (FMR16xG and FMR99xG, respectively). We also silenced IGF2BP3 in previously generated HEK293 FlipIn T-Rex cell models with a stable expression of FMRpolyG containing either 16 or 95 glycine residues (S-FMR16xG and S-FMR95xG lines, respectively). In all of the tested cell models, including transient and stable expression systems, we observed a significant, approximately two-fold, reduction of FMR99xG protein levels upon IGF2BP3 KD (Fig. 2A, Supplementary Fig. 2A-C). Importantly, we also observed (although to a lower extent (20−25%)), a decrease in the shorter protein, FMR16xG, upon IGF2BP3 KD (Fig. 2A and Supplementary Fig. 2A), suggesting that IGF2BP3 regulates RAN-translated FMRpolyG synthetized from both normal and PM CGG length thresholds. Additionally, in tested cell models, we did not observe an effect of IGF2BP3 KD on endogenous FMRP levels, which is produced from the same mRNA as FMRpolyG, but different open reading frame (Fig. 2A). On the other hand, IGF2BP3 KD led to an increase of endogenous *FMR1* mRNA level (Fig. 2A), suggesting that *FMR1* is a target of IGF2BP3, which is in line with eCLIP data (Supplementary Fig. 1G).

Since IGF2BP3 plays a role in various aspects of RNA metabolism, we also investigated the effects of IGF2BP3 KD on *FMR1* transcripts containing either 16 or 99 CGG repeats. The quantitative real-time polymerase chain reaction (RT-qPCR) analysis showed that, in a stable expression system, the mRNA levels of *FMR16xG* and *FMR95xG* increased approximately 2-fold (Fig. 2A). To our surprise, in the transient expression system, the relative *FMR99xG* mRNA levels increased 2.5-fold in HEK293T cells, while the *FMR16xG* mRNA with a short repeat tract remained unchanged (Supplementary Fig. 2A).

We also performed nuclear-cytoplasmic fractionation to verify whether the transport of *FMR99xG* mRNA is impaired upon IGF2BP3 KD. In both nuclear and cytoplasmic fractions isolated from the FMR99xG cell model, we observed a significant increase of *FMR99xG* mRNA, with a similar trend as in *FMR99xG* measured for the total RNA (Fig. 2B). Together, these data suggest that the decrease in FMRpolyG

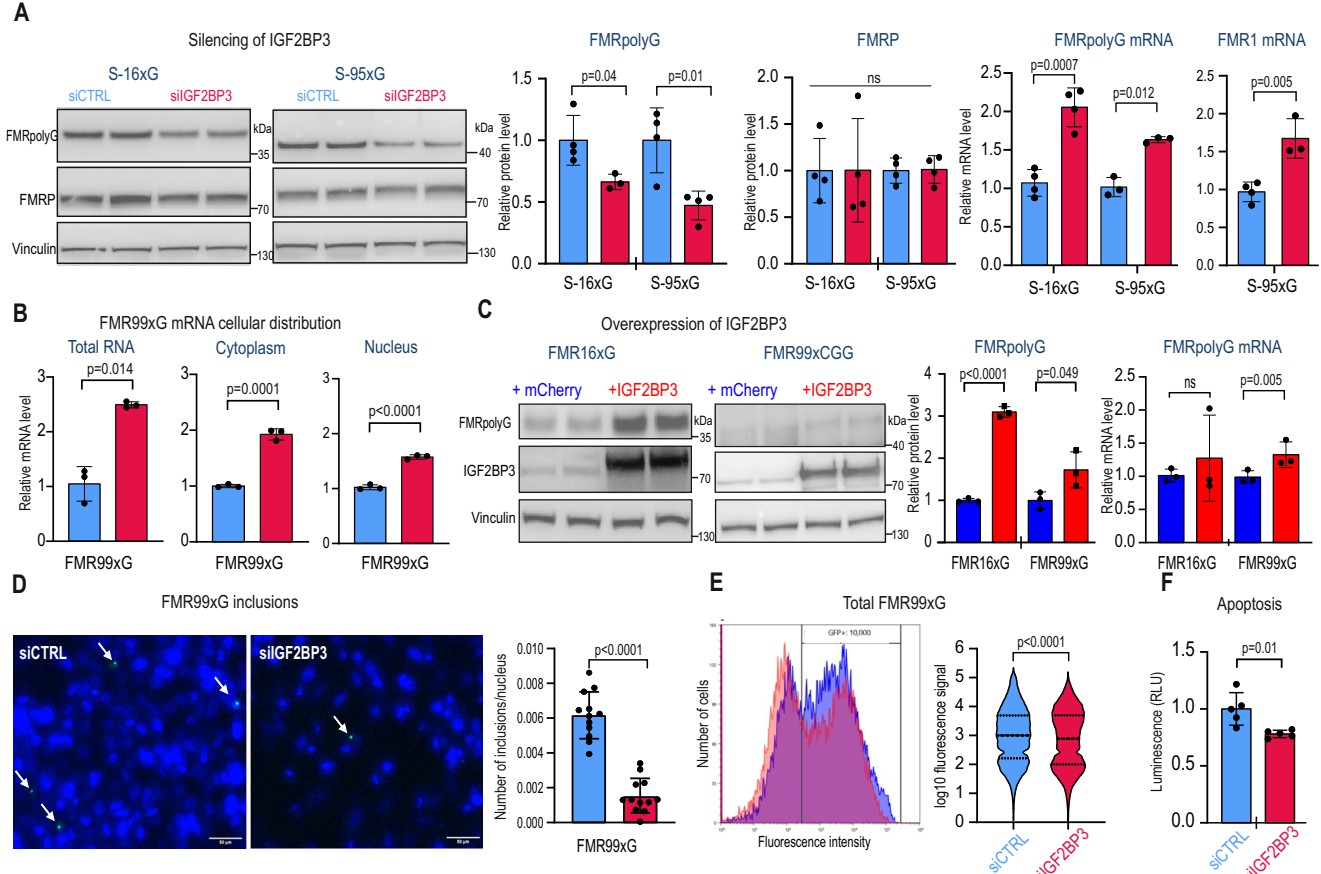

**Fig. 2 | IGF2BP3 regulates the level of repeat-associated non-AUG initiated (RAN)-translated FMRpolyG in different cell models. A** The effect of IGF2BP3 knockdown (KD, magenta) on FMRpolyG proteins containing a long (S-FMRx95G) or short (S-FMRx16G) polyglycine stretch, and canonical FMRP protein in two cell models with stable expression of FMRpolyG, measured by Western blot (WB) and normalized to Vinculin (left). The effect of IGF2BP3 silencing on *FMR16xG* and *FMR95xG* transgene and *FMR1* endogene expression quantified with RT-qPCR and normalized to *GAPDH* (right). *N* = 4 biologically independent samples. **B** The effect of IGF2BP3 KDs (magenta) on the cellular distribution of *FMR99xG* transgene mRNA measured with RT-qPCR for *N* = 3 biologically independent samples. The results of total and cytoplasmic fractions were normalized to *GAPDH* and the results of nuclear fractions were normalized to *MALAT*. **C** The effect of IGF2BP3 over-expression (red) on FMRpolyG protein levels in a transient transfection system, measured by WB and normalized to Vinculin (left). The effect of IGF2BP3 over-expression on *FMR16xG* and *FMR95xG* transgene expression quantified by RT-qPCR and normalized to *GAPDH*. *N* = 3 biologically independent samples. **D** Microscopic

quantification of FMRpolyG-GFP inclusions (white arrows) in HeLa cells transfected with the *FMR99xG* construct on the background of an IGF2BP3 KD (magenta). Representative images were pseudo-colored and merged; green, GFP-positive inclusions; blue, nuclei stained with Hoechst 33342; scale bars, 50 μm. The barplot presents mean number of FMRpolyG-GFP inclusions per nucleus in *N* = 6 biologically independent samples with SDs. (**E**) Cytometric quantification of the total FMRx99G in HEK293T cells expressing the *FMR99xG* on the background of an IGF2BP3 KD. The fluorescence signal of GFP-positive cells was measured, excluding dead cell populations. The violin plots present different FMRpolyG-GFP signal distributions in cells treated with siCtrl (blue) and siIGF2BP3 (magenta). *N* = 4 biologically independent samples. **F** The effect of IGF2BP3 KDs on apoptosis evoked by FMRpolyG expression in HEK293T cells. Apoptosis was measured as luminescence signals (relative luminescence units; RLU). *N* = 6 biologically independent samples. **A–F** An unpaired two-sided t-test was used to calculate statistical significance: ns, non-significant. Data are shown as means with standard deviations.

level is not a result of a reduced RNA level, impaired nuclear retention, or defective nucleocytoplasmic transport.

Importantly, we found that overexpression of IGF2BP3 induced opposite effects on FMRpolyG levels than KDs (Fig. 2C). Co-transfection of HEK293T cells with an *IGF2BP3*-overexpressing construct and *FMR16xG-* or *FMR99xG*-constructs led to a 3-fold and 1.7-fold increase of FMR16xG and FMR99xG protein levels, respectively. IGF2BP3 overexpression did not affect *FMR16xG* mRNA levels, however, it slightly increased the *FMR99xG* mRNA level by 1.3-fold.

Toxic FMRpolyG proteins can be present in cells in soluble, insoluble, and aggregated form, therefore, we investigated whether both protein states would be affected by IGF2BP3 depletion. We monitored the number of GFP-positive FMRpolyG aggregates in HeLa cells and found that their number was significantly decreased upon IGF2BP3 KD (Fig. 2D). This finding was further confirmed using a flow

cytometry assay, where the average fluorescence signal of FMRpolyG-GFP, emitted by both soluble and aggregated forms of FMR99xG, was significantly decreased in HEK293T cells with lowered IGF2BP3 expression (Fig. 2E). Moreover, we found that apoptosis evoked by toxic FMR99xG was also significantly diminished in the same cell model (Fig. 2F).

### IGF2BP3-dependent regulation is specific to the RAN translation of FMRpolyG and dependent on the *FMR1* RNA sequence context

Next, we investigated whether IGF2BP3-dependent regulation is specific to noncanonical RAN biosynthesis, characterized by initiation at non-AUG codons. Therefore, we tested the effect of IGF2BP3 KDs on the FMRpolyG produced from mRNA in which the near-cognate ACG codon was replaced by the AUG codon (*ATG-FMR99xG*). As expected, this substitution significantly increased the basal FMRpolyG level. However, the protein level of AUG-FMR99xG derived from the AUG

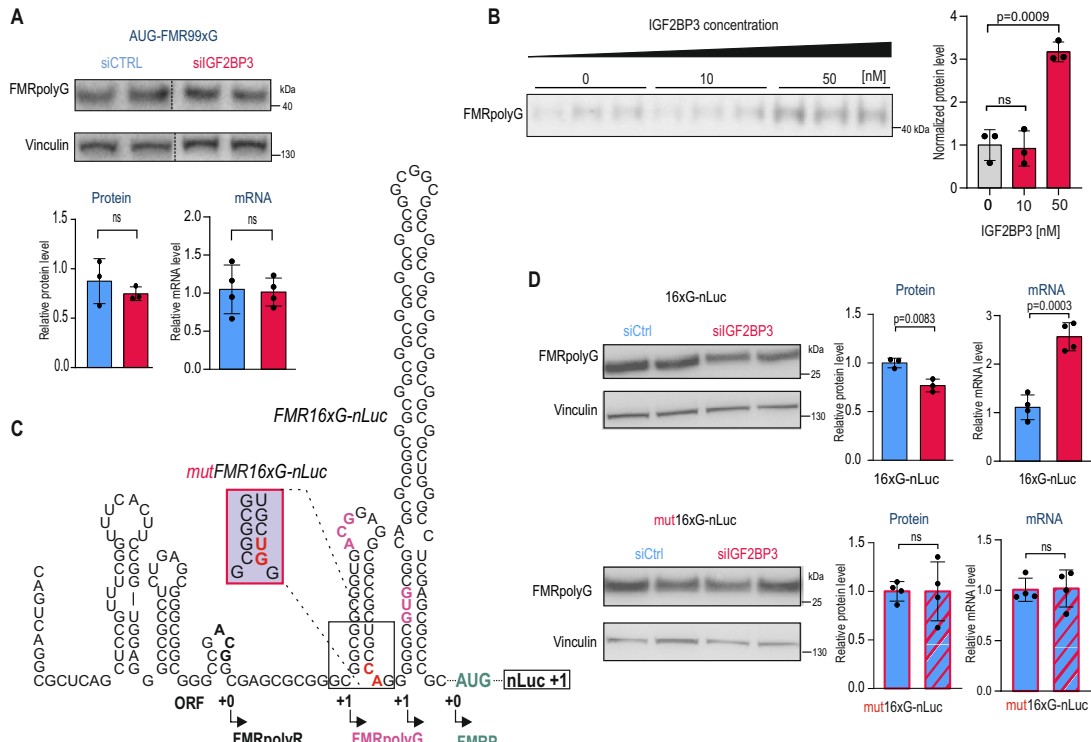

**Fig. 3 | Effect of IGF2BP3 on FMRpolyG is specific to repeat-associated non-AUG initiated (RAN)-translation. A** The effect of IGF2BP3 knock down (KDs, red) on FMRx99G translated from the canonical AUG codon (AUG-FMRx99G) measured by Western blot (WB) and normalized to Vinculin (left). The effect of an IGF2BP3 KD on *AUG-FMR1-GFP* transgene expression was quantified with RT-qPCR and normalized to *GAPDH* (right). *N* = 3 biologically independent samples. **B** WB analysis of FMR16xG produced via in vitro translation using rabbit reticulocyte lysate in the absence or presence of increasing concentrations of recombinant IGF2BP3 (0, 10, and 50 nM). *N* = 3 independent experiments per IGF2BP3 concentration. **C** The predicted secondary structure of *FMR16xG-nLuc* RNA, highlighting the potential CA motif required for IGF2BP3 binding, mutagenesis of the CA motif to UG, and its effect on predicted *mutFMR16xG-nLuc* RNA secondary structure is shown in red

rectangle. **D** The effect of an IGF2BP3 KD (red) on the FMR16xG protein fused with nanoluciferase reporter (16xG-nLuc) measured by WB and normalized to Vinculin (left, upper panel). The effect of an IGF2BP3 KD on *16xG-nLuc* transgene expression quantified with RT-qPCR and normalized to *GAPDH* (right, upper panel). The effect of an IGF2BP3 KD on the level of FMR16xG-nanoluciferase fusion protein produced from *mutFMR16xG-nLuc* mRNA with a mutated CA motif (mut16xG-nLuc) measured by WB and normalized to Vinculin (left, lower panel). The effect of an IGF2BP3 KD on *mut16xG-nLuc* transgene expression quantified with RT-qPCR and normalized to *GAPDH* (right, lower panel). *N* = 4 biologically independent samples. (**A**, **B**, **D**) An unpaired two-sided t-test was used to calculate statistical significance: ns, non-significant. Data are shown on the graphs as means with standard deviations. Presented gels were cropped.

codon, and its mRNA level, remained unchanged upon IGF2BP3 KD (Fig. 3A).

We then tested whether supplementation of an in vitro translation reaction with recombinant IGF2BP3 affects the biosynthesis of FMRpolyG. We used cell-free rabbit reticulocyte lysate to translate in vitro RNA encoding the 5'UTR sequence of the *FMR1* gene with 16 CGG repeats. In accordance with previous studies[25], a WB analysis confirmed the efficient production of FMRpolyG. Importantly, the addition of recombinant IGF2BP3 to in vitro transcribed *FMR16xG* RNA at a 2:1 molar ratio significantly enhanced FMR16xG synthesis (Fig. 3B). This finding confirms the direct effect of IGF2BP3 on RAN translation.

Recently, an integrative study revealed that the IGF2BP3 protein recognizes RNA containing CA- motifs and GGC core elements (either GGCA or CGGC) separated by appropriate spacing[38]. Since multiple CA motifs are located upstream of CGG repeats in the 5'UTR of *FMR1*, we investigated whether one of these CA dinucleotides is critical for IGF2BP3 binding. First, we overexpressed the construct encoding the 5'UTR region of *FMR1* containing 16 CGG repeats fused with nanoluciferase (*FMR16xG-nLuc*) in HEK293T cells, and found that depletion of IGF2BP3 in this model resulted in decreased FMR16xG protein levels (Fig. 3D). Of note, the *FMR16xG-nLuc* transcript level was upregulated upon IGF2BP3 silencing, similar to the *FMR16xG* transcript produced from the construct with the GFP tag. This experiment confirmed that the type of tag used in transgene generation does not affect the

sensitivity of tested RNA and proteins to the amount of IGF2BP3. Then, using mutagenesis, we replaced CA with the TG dinucleotide motif located 20 nucleotides upstream of the CGG repeats and 14 nucleotides downstream of the ACG near cognate start codon (*mutFMR16xG-nLuc*, Fig. 3C). We found that this mutation did not impede production of FMR16xG, but the mutant FMR16xG and its mRNA level were not sensitive to IGF2BP3 silencing (Fig. 3D). This indicates that the CA dinucleotide is crucial for IGF2BP3 binding, and mutation of this site abolishes the regulatory role of this protein on the RAN translation of *FMR1* mRNA.

These results suggest that the IGF2BP3 protein controls FMRpolyG protein levels, independent of CGG repeat length, and that the regulation is specific to RAN but not canonical translation, as no impact on AUG-translated FMRpolyG nor endogenous FMRP proteins was observed upon IGF2BP3 silencing. IGF2BP3 modulates the RAN translation of CGG repeats via direct interaction with the 5'UTR of normal and mutant *FMR1* mRNA.

### Other IGF2BP paralogs regulate the biosynthesis of RAN-translated FMRpolyG

The IGF2BP family is composed of three members: IGF2BP1, IGF2BP2, and IGF2BP3, which are characterized by high sequence identity (Supplementary Fig. 4E) and similar composition of RNA binding domains[32,39,40]. IGF2BP paralogs govern a different pool of cellular

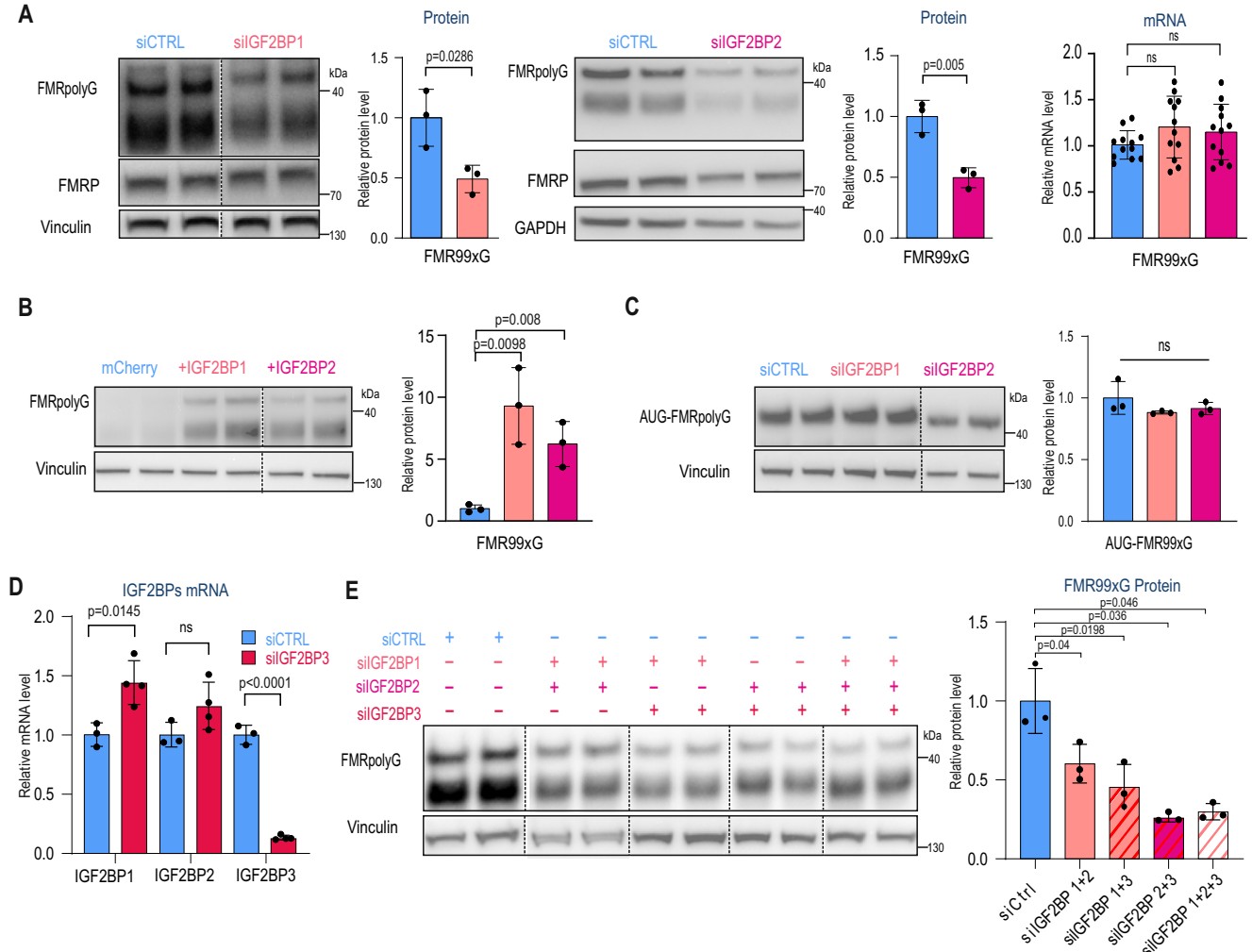

**Fig. 4 | The level of IGF2BP paralogs influences FMRpolyG biosynthesis. A** The effect of IGF2BP1 knockdowns (KDs, coral) on FMRpolyG protein containing a long polyglycine stretch (FMR99xG) and canonical FMRP protein as measured by Western blot (WB) and normalized to Vinculin (left). The effect of an IGF2BP2 KD (magenta) on the FMRpolyG protein containing a long polyglycine stretch (FMR99xG) and canonical FMRP protein as measured by WB and normalized to Vinculin (middle). The effect of IGF2BP1 and IGF2BP2 silencing on *FMR99xG* transgene expression quantified with RT-qPCR and normalized to *GAPDH* (right). **B** The effect of IGF2BP1 (coral) and IGF2BP2 (magenta) overexpression on the FMRpolyG protein containing a long polyglycine stretch (FMR99xG) measured by Western blot and normalized to Vinculin. The overexpression of mCherry was used as a control. **C** The effect of IGF2BP1 (coral) and IGF2BP2 (magenta) KDs on the FMRpolyG protein translated from the canonical AUG codon (*ATG-FMRx99G*) measured by WB and normalized to Vinculin. **D** The effect of IGF2BP3 silencing (red) on IGF2BP1, IGF2BP2, and IGF2BP3 gene expression quantified with RT-qPCR and normalized to *GAPDH*. **E** The effect of simultaneous silencing of IGF2BP1, IGF2BP2, and IGF2BP3 on the FMRpolyG protein containing a long polyglycine stretch (FMR99xG) measured by WB and normalized to Vinculin. **A–E** The graphs present means from at least *N* = 3 biologically independent samples with standard deviations (SDs). An unpaired two-sided t-test was used to calculate statistical significance: ns, non-significant. Presented gels were cropped.

RNAs by multiple mechanisms, including sequestration in mRNP granules and the protection or enhancement of RNA degradation (reviewed in ref. 41), and the binding specificity is determined by differences in amino acid sequence in variable loops of KH domains[40]. Importantly, some of the target RNA overlap, such as *CD44*, *CRC*, *MYC* and *TNBC* (reviewed in Duan et al.[42]). Additionally, IGF2BP1 was previously reported, by us and others, as protein binding to the *FMR1* 5′UTR containing expanded CGG repeats *in cellulo*[36,43], or CGG repeats without *FMR1* sequence context[44]. Thus, we aimed to check whether IGF2BP1 and IGF2BP2 paralogs can regulate levels of FMR99xG similarly to IGF2BP3. We found that silencing IGF2BP1 and IGF2BP2 resulted in a 2-fold decrease of transiently expressed FMR99xG protein levels, without impairing its mRNA level (Fig. 4A), and co-expression of FMR99xG with IGF2BP1 or IGF2BP2 resulted in 9- and 6-fold increase in FMR99xG protein levels, respectively (Fig. 4B). Additionally, we found that KDs of IGF2BP1 or IGF2BP2 paralogs did not influence the level of FMRpolyG translated from a canonical AUG codon, similar to that of

the IGF2BP3 paralog (Fig. 4C). We then asked whether IGF2BP1 and IGF2BP2 can compensate for IGF2BP3 level and found that only IGF2BP1 mRNA level increased upon IGF2BP3 KD (Fig. 4D). However, combined silencing of three paralogs revealed that IGF2BP2 and IGF2BP3 contribute most significantly to the regulation of FMRpolyG level in HEK293T cells (Fig. 4E). Together, this data suggests that all IGF2BP paralogs specifically affect the level of RAN translated FMRpolyG containing long polyG tracts.

**Small molecule inhibitors targeting IGF2BP3 modulate the level of FMRpolyG**

Several preclinical studies have demonstrated successful down-regulation of IGF2BP3 expression using different small compounds, including isocorydine derivative[45] and bromo- and extra-terminal domain (BET) inhibitors such as JQ1[46–48] and I-BET151[49,50] (Fig. 5A). Here, we aimed to verify whether these small molecules can modulate the level of FMR99xG through the inhibition of IGF2BP3. We administered

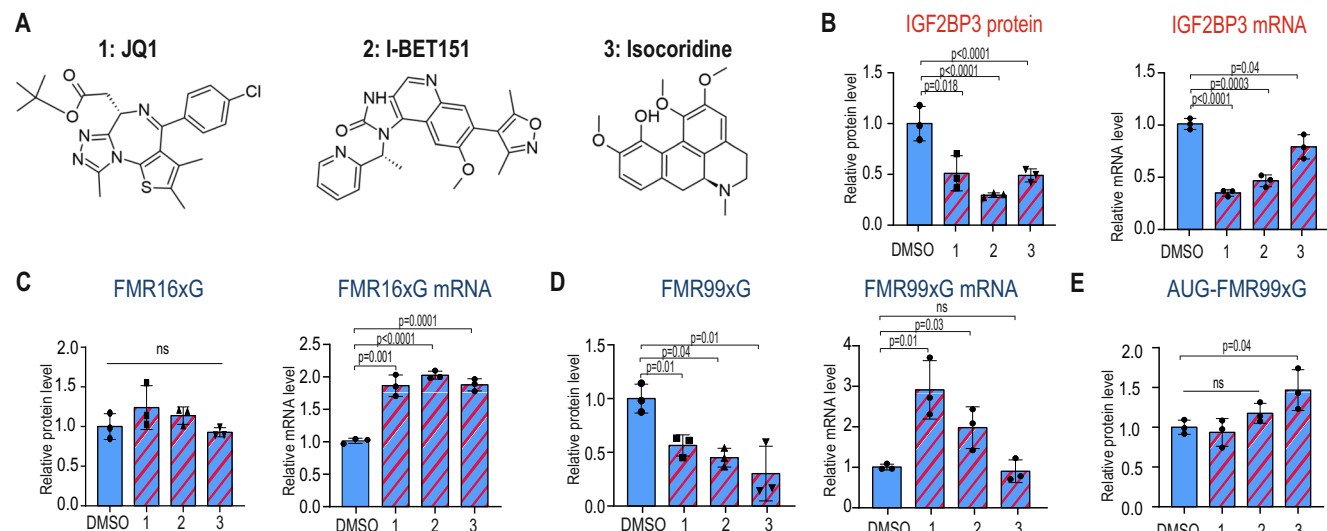

**Fig. 5 | Small molecule inhibitors targeting IGF2BP3 induce decreases in FMRpolyG levels. A** Chemical structure of JQ1 (1), I-BET151 (2), and Isocoridine (3) molecules used in the study. **B** The effect of JQ1 (1), I-BET151 (2), and Isocoridine (3) on endogenous IGF2BP3 protein levels in HEK293T cells measured by Western blot (WB) and normalized to Vinculin (left) and on endogenous *IGF2BP3* mRNA levels (right) measured by RT-qPCR and normalized to *GAPDH*. **C** The effect of administering (1), (2), and (3) on the levels of FMRpolyG containing a short polyglycine stretch (FMRx16G) measured by WB and normalized to Vinculin (left) and *FMR16xG* transgene expression measured by RT-qPCR and normalized to *GAPDH*. HEK293T cells were transfected with construct-encoding, short FMR16xG-GFP. **D** As in C but for HEK293T cells transfected with construct-encoding, long FMR99xG-GFP. **E** As in C but for HEK293T cells transfected with the *ATG-FMR99xG-GFP* construct with substitution of the ACG to the AUG initiation codon for long FMRpolyG. **B–E** The graphs present means from *N* = 3 biologically independent samples with standard deviations. An unpaired two-sided t-test was used to calculate statistical significance: ns, non-significant.

these compounds to HEK293T cells with transient expressions of FMR16xG or FMR99xG and found that, although all three molecules induced an efficient decrease of IGF2BP3 protein levels in line with previous reports, only JQ1 and I-BET151 decreased the *IGF2BP3* mRNA level (Fig. 5B). This suggests that isocorydine inhibits the IGF2BP3 level via a different mechanism than BET inhibitors. Importantly, cells treated with all of the tested inhibitors significantly reduced FMR99xG but not FMR16xG protein levels (Fig. 5C, D). Moreover, both JQ1 and I-BET151 induced the upregulation of mRNA levels of both *FMR16xG* and *FMR99xG*, similar to the knockdown of IGF2BP3 (Fig. 5C, D). The actions of JQ1 and I-BET151 appear to be specific to RAN-translated FMR99xG, as administration of these molecules to HEK293T cells with transient expressions of *ATG-FMR99xG* did not affect the level of FMRpolyG produced from the canonical start codon (Fig. 5E). Additionally, only JQ1 significantly decreased FMRP protein level, while both JQ1 and I-BET151 decreased *FMR1* mRNA level (Supplementary Fig. 5B, D). We have also found that administration of JQ1 decreased the relative mRNA level of *IGF2BP1* and *IGF2BP2* (Supplementary Fig. 5D), which is in line with previous findings[51].

In sum, we conclude that the tested small compounds, which have well-described anticancer activities, significantly reduced the biosynthesis of toxic FMR99xG similar to IGF2BP3 KDs, which suggests a therapeutic potential for these and similar molecules in FXPAC.

### IGF2BP3 regulates the level of *FMR1* mRNA in FXTAS patient derived cells

To validate our findings in an endogenous context, we silenced IGF2BP3 in control and FXTAS patient-derived fibroblasts and found that the relative *FMR1* mRNA level increases upon IGF2BP3 KD (Fig. 6A, Supplementary Fig. 6A), similarly as in cell lines with stable expression of FMRpolyG (Fig. 2A). Furthermore, we differentiated two control and two FXTAS-patient derived induced pluripotent stem cells (iPSCs), into neural progenitor cells (NPCs, Supplementary Fig. 6E). We found that silencing of IGF2BP3 in this model resulted in increased relative *FMR1* mRNA levels in FXTAS NPC (Supplementary

Fig. 6B), similarly as observed in fibroblasts (Fig. 6A). The NPCs were then differentiated into neurons, and after two weeks of culture, cells were transfected with either a non-targeting control siRNA or IGF2BP3 siRNA. In this model, we also observed the increase of relative *FMR1* mRNA level upon IGF2BP3 KD (Fig. 6B, Supplementary Fig. 6C), confirming our observations in overexpression models (Fig. 2A, Fig. 6A). Expression of CGG RAN translation products was previously shown to decrease cell viability[52] and induce neuronal cell death[17,18]. In line with these findings, we measured the impact of IGF2BP3 silencing on necrosis progression in control and FXTAS iPSC-derived neurons, starting at forty-eight hours post siRNA treatment. We found that lower IGF2BP3 expression rescued the necrosis phenotype in FXTAS iPSC-derived neurons (Fig. 6C), but not in control cells (Supplementary Fig. 6F). After three weeks of neuronal differentiation, using immunofluorescence, we detected FMRpolyG in 5 to 10% of FXTAS iPSC-derived neurons (Fig. 6D, Supplementary Fig. 6G), but not in controls (Supplementary Fig. 6H), in agreement with previous reports[17]. We also found that IGF2BP3 KD in these cells resulted in a modest, but not statistically significant reduction in the number of FMRpolyG aggregates (Fig. 6D). We suppose that this can be due to lower IGF2BP3 KD efficiency (Supplementary Fig. 6D), which was not sufficient to inhibit FMRpolyG production and aggregate formation. Finally, we administered BET inhibitors: JQ1 and I-BET151, to control and FXTAS iPSC-derived neurons and measured apoptosis rate after 6 days of treatment. We observed that both JQ1 and I-BET151 decreased the apoptosis rate in FXTAS iPSC-derived neurons, but not in controls (Supplementary Fig. 6I).

### imph-1/IGF2BP disruption rescues 99xCGG phenotype in *Caenorhabditis elegans*

To determine whether the effects of IGF2BP knockdown on the FXPAC phenotype could be replicated in an animal model, we generated transgenic *C. elegans* expressing RNA of human *FMR1* 5'UTR with 99 CGG repeats, fused to GFP at the 3' end *(polyG(agaEx7[99xCGG]))*,

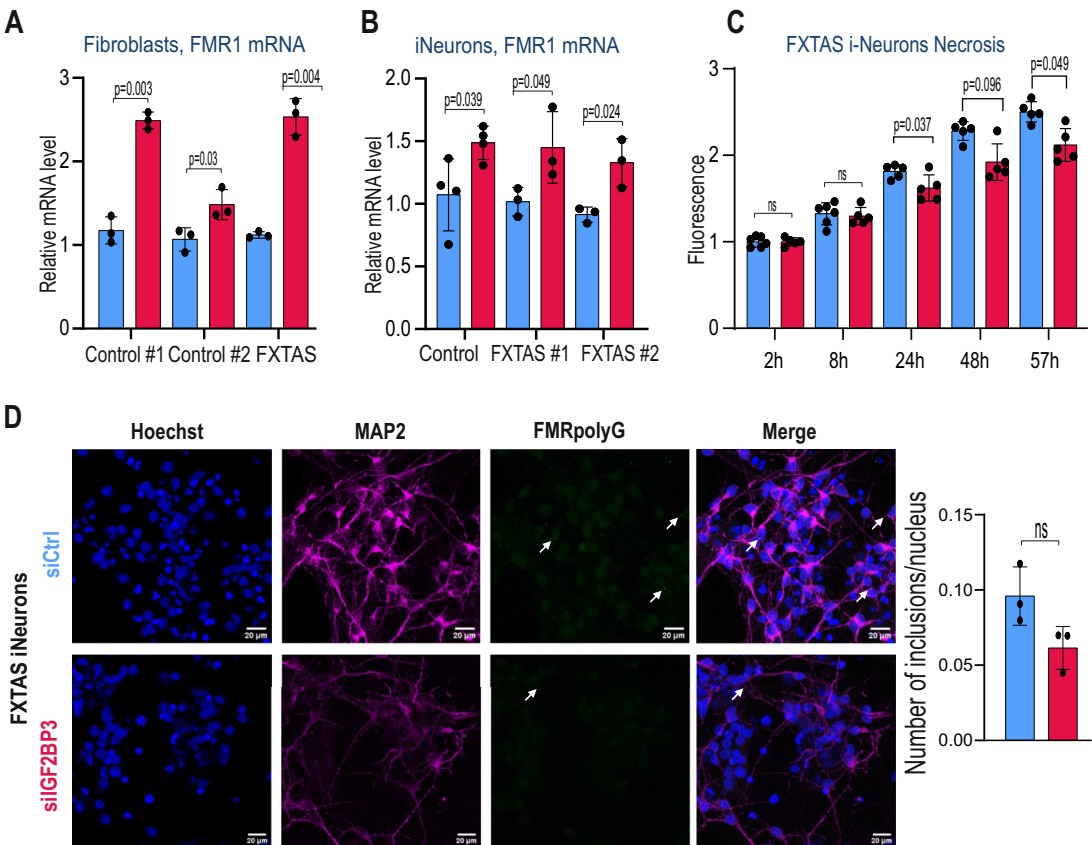

**Fig. 6 | IGF2BP3 influences FMR1 mRNA level and rescues cell death in FXTAS patient derived cells. A** The effect of IGF2BP3 silencing on FMR1 expression in control (22 CGG, 31 CGG) and FXTAS (81 CGG) fibroblasts, quantified with RT-qPCR and normalized to GAPDH. The graphs present means from *N* = 3 biologically independent samples with standard deviations (SDs). **B** The effect of IGF2BP3 silencing on FMR1 expression in control (29CGG) and FXTAS (#1 83CGG, #2 72CGG) iPSC-derived neurons, quantified with RT-qPCR and normalized to *GAPDH*. The graphs present means from *N* = 4 biologically independent samples with SDs. **C** IGF2BP3 silencing lowers necrosis rate in FXTAS (72 CGG) iPSC-derived neurons. Necrosis was measured as fluorescence signals (relative fluorescence units). The graph presents relative mean values from *N* = 5 biologically independent samples treated with siCtrl (blue) or siIGF2BP3 (red), with SDs. **D** Effect of IGF2BP3 silencing on FMRpolyG inclusions in FXTAS (83 CGG) iPSC-derived neurons. Immunofluorescence against FMRpolyG N-terminus (white arrows) was performed on neuronal cultures differentiated 21 days from FXTAS iPSC (left) on the background of IGF2BP3 silencing. The histogram (right) presents a mean number of FMRpolyG inclusions per nucleus in cells treated with siCtrl (blue) or siIGF2BP3 (red), in *N* = 3 independent cultures with SDs. Representative images were pseudo-colored and merged; green, FMRpolyG-positive inclusions; purple, neuronal marker MAP2; blue, nuclei stained with Hoechst 33342; scale bars, 20 μm. Minimum 100 neurons per micrograph were used for quantification. **A**–**D** An unpaired two-sided t-test was used to calculate statistical significance: ns non-significant.

herein referred to as 99xCGG model (Fig. 7A). Microscopy analysis revealed that these worms expressed low level of FMRpolyG-GFP in the head (Fig. 7B, Supplementary Fig. 7A), which was further confirmed by fluorescence lifetime imaging (FLIM), showing a GFP lifetime of 2.49 ns (Supplementary Fig. 7A). We then sought to assess the role of *C. elegans* IGF2BP ortholog, *imph-1*, which shares almost 30% amino acid sequence with human IGF2BP paralogs (Supplementary Fig. 7B), in the 99xCGG *C. elegans* model. To do so, we crossed the 99xCGG transgenic strain with the *imph-1(sy1993)* mutant animals, which carries a stop-in cassette insertion disrupting *imph-1* expression. In this 99xCGG; *imph-1Δ* genetic background, we observed a decrease of FMRpolyG-GFP fluorescence (Fig. 7B, C), and increase in FMR1-99CGG mRNA levels compared to the 99xCGG strain alone (Fig. 7D), consistent with the upregulation seen in FXTAS fibroblast and FXTAS iPSC-derived neurons (Fig. 6A, B).

Furthermore, we observed that 99xCGG animals exhibited severe phenotypic defects, including reduced brood size, impaired thrashing behavior and decreased lifespan, relative to wild-type animals (Fig. 7E–G, Supplementary Movies 1–4). Notably, these phenotypes were significantly rescued in the 99xCGG; *imph-1Δ* background, suggesting that the product of *imph-1* positively regulates FMRpolyG expression and that its loss may mitigate the detrimental effects of CGG repeat expansion in model organism.

## Discussion

Noncanonical RAN translation leading to the production of aberrant proteins, was initially described in spinocerebellar ataxia type 8 (SCA8)[53] and, to date, has been reported to be involved in at least nine disorders caused by RNA repeat expansion, including FXTAS[16,17], amyotrophic lateral sclerosis, and frontotemporal dementia (ALS/FTD) caused by expansion of GGGGCC repeats in the *C9orf72* gene[54]. RAN products have been shown to accumulate in patients' tissues, and emerging evidence suggests that they may be involved in disease pathology (reviewed in Banez-Coronel & Ranum[55]). Considerable effort has been made to decipher the mechanism of RAN translation and identify trans-acting factors regulating this process (reviewed in Baud et al.[56]). Nevertheless, while numerous studies have shed light on the RAN translation mechanism, many aspects of this process remain unclear as the mechanism may differ depending on the RNA repeat type. For instance, RAN translation of CGG repeats may be similar to upstream open-reading frames[57] as its initiation is 5'-cap-, eIF4A-, and eIF4E-dependent[25], while GGGGCC repeat-specific RAN translation is cap-independent and initiates similarly to the internal ribosomal entry site (IRES)[58]. In parallel, some RAN translation modifiers may have opposite effects on RAN products, notably the depletion of DDX3X helicase upregulated GGGGCC-RAN translation[59], while a KD of a DDX3X homolog in a *Drosophila* FXTAS model was shown to inhibit

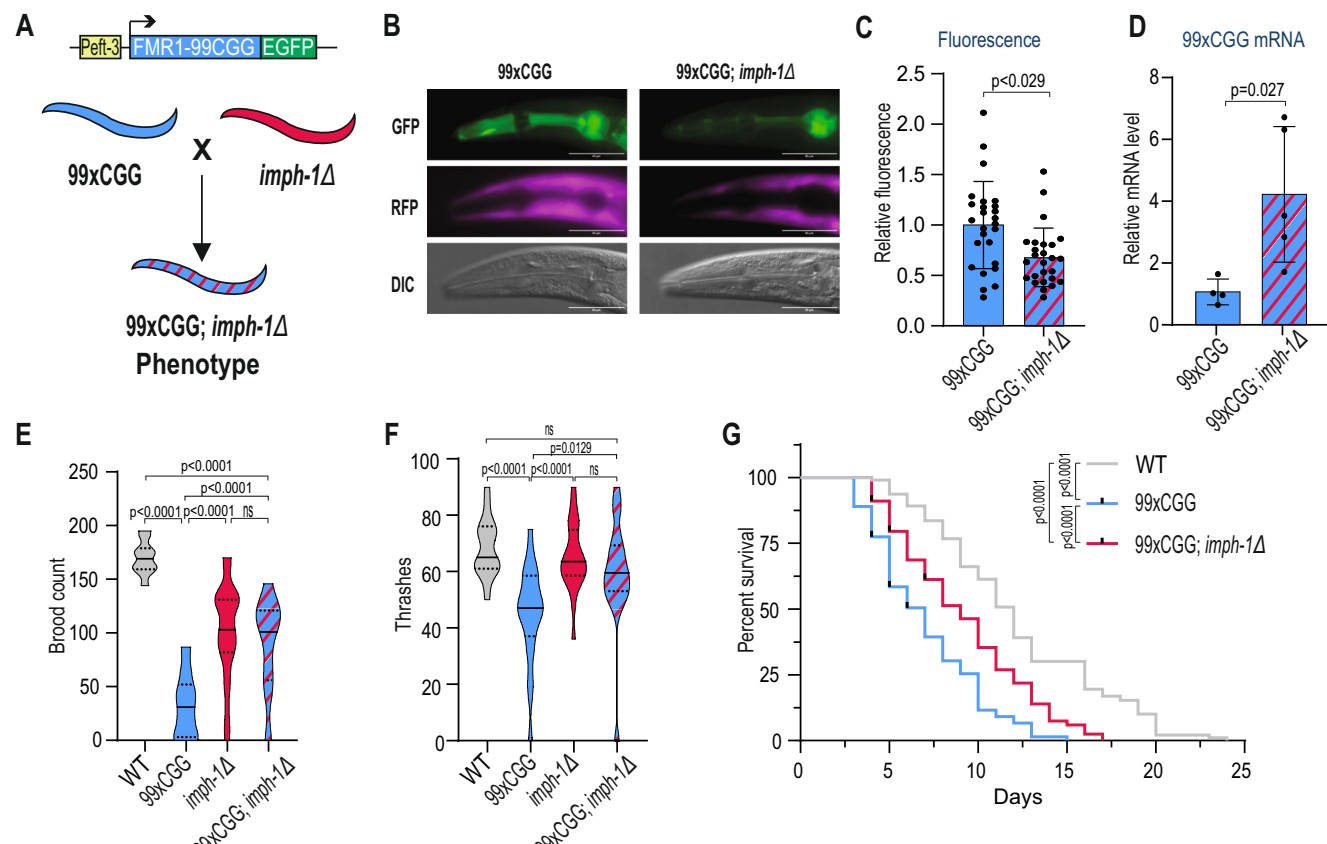

**Fig. 7 | Functional knock-out of *imph-1* mitigates CGG repeat-associated toxicity in *C. elegans*. A** Schematic of the construct used to generate FXTAS *C. elegans* model (upper panel) and experimental outline for phenotype screening. **B** Representative images of 99xCGG and 99xCGG; *imph-1Δ* animals. Images were pseudo-colored: green, FMRpolyG-GFP; purple, Red Fluorescence Protein; DIC-differential interference contrast, scale bars, 50 μm. **C** Confocal microscopy quantification of FMRpolyG-GFP fluorescence signal in 99xCGG and 99xCGG; imph-1Δ animals. Each animal is represened by a single data point. The graphs present means from *N* = 26 biologically independent samples with standard deviations (SDs). **D** RT-qPCR quantification of FMR1-99CGG-GFP RNA extracted from animals of indicated phenotype and normalized to actin. The graphs present means from

*N* = 4 biologically independent samples with standard deviations (SDs). **E** Violin plots of the brood count (viable progeny) assay of *C.elegans* wild type (WT), 99xCGG, imph-1Δ and 99xCGG; imph-1Δ genotypes. Observation based on *N* = 10 biologically independent samples for WT genotype and *N* = 20 for remaining genotypes. **F** Violin plots depicting the locomotion (thrashing assay) of indicated animals. Sample sizes (N) for each genotype are as follows: WT: 19, 99xCGG: 21, imph-1Δ: 22, and 99xCGG; imph-1Δ: 20. **G** Survival curve of *C.elegans* WT (*N* = 212), 99xCGG (*N* = 218), and 99xCGG; imph-1Δ (*N* = 213) genotypes. Mantel-Cox test was used to calculate statistical significance of survival curves. **C–E** An unpaired two-tailed t-test was used to calculate statistical significance: ns non-significant.

CGG-RAN translation[29]. Other factors, such as DHX36 helicase, may have an analogous impact on the production of RAN proteins from different repeats[30]. The identification of factors governing RAN translation can provide insight into its mechanism and indicate potential therapeutic targets that could be used to treat incurable diseases caused by RNA repeat expansion.

Several studies have attempted to identify rCGGexp binding proteins and modifiers of CGG-RAN translation using distinct methods. Proteomic approaches were successfully used to identify proteins binding to various lengths of rCGG repeats[11–13]. However, these in vitro experiments were based on constructs devoid of the natural 5′UTR of the *FMR1* gene and, thus, focused primarily on protein sequestration by rCGGexp. Whole genome sequencing combined with genetic screening has revealed proteasome subunit beta-type 5[60], the component of the core 20S proteasome complex, as a modifier of RAN translation. Candidate-based screens have deciphered the role of the DDX3X helicase in unwinding RNA secondary structures of mutant *FMR1* RNA, thus facilitating PIC scanning[29] and alternative ternary complex factors in selectively modulating RAN translation[61]. However, these screens, based on a priori hypotheses, likely disregarded some potential modifiers. Finally, we, and others, have used RNA-tagging

systems combined with proteomic analysis to capture proteins interacting with mutant 5′UTRs of *FMR1* RNA in cellulo[36,43]. This approach revealed that two ribosomal proteins, RPS26 and RPS25, positively regulate the translation of FMRpolyG[36]. In another screen, serine and arginine-rich splicing factor 1, which binds to mutant *FMR1* RNA and affects its nucleocytoplasmic transport, was identified[43].

In this work, we aimed to determine the proteins interacting with *FMR1* 5′UTR RNA within the normal or PM CGG repeat range. Using in vitro RNA pull-down combined with MS, we identified 80 enriched proteins binding to FMR1-99CGG RNA. Importantly, although in vitro approaches have some limitations due to possible non-specific interactions[62], in our screen we identified proteins that were previously described based on *in cellulo* assays conducted by us[36] and others[11,13], confirming the credibility of our data. Additionally, we identified a novel set of proteins binding to the 5′UTR-*FMR1* devoid of repeats, which may play a role in regulating translation efficiency in physiological conditions.

Downstream validations of identified hits revealed IGF2BP3 as a specific regulator of CGGexp-RAN translation in different FXTAS cell models, FXTAS patient-derived cells and animal model expressing fragment of mutant *FMR1*. This multi-domain RNA binding protein can

associate with a plethora of targets and influence RNA at different levels (reviewed in ref. 39). IGF2BP3 has been shown to promote the translation of *IGF2* mRNA[35], protect the cyclins D1, D3, and G1 mRNAs from translational repression[63], and partition mRNA between translating and non-translating pools from polysomes to P-bodies[64]. Other members of the IGF2BP family include IGF2BP1 and IGF2BP2, which share high sequence identity and domain composition with IGF2BP3[32], have multiple RNA targets, and regulate RNA fate in different processes (reviewed in Bell[39]). For instance, IGF2BP1 binds the 3′UTRs of *ACTB* and *MAPK4* mRNAs and inhibits their translation[65,66], while IGF2BP2 enhances *IGF2* mRNA translation[67]. Importantly, IGF2BP1 was previously identified as protein binding to the *FMR1* 5′UTR containing CGGexp[36,43,44]. In this study, we found that all IGF2BP paralogs regulate protein levels of RAN-translated FMR99xG, while not impairing the levels of FMRpolyG translated from the canonical AUG codon and canonical FMRP, which is produced from different open reading frame of the same mRNA. This may be explained by the fact that the AUG-FMRpolyG and FMRP translation starts in the most favorable Kozak context, leading to the optimal protein biosynthesis. Therefore, the presence of additional factors, such as IGF2BPs will not change FMRpolyG level. Silencing of IGF2BP1 and IGF2BP2 did not impair *FMR99xG* mRNA levels, in contrast to IGF2BP3. Thus, we cannot exclude that the mechanism of action of each paralog is at least partially different. Notably, IGF2BP paralogs are expressed in different tissues[68], which should be taken into account when designing potential therapies targeting these proteins. This is especially important as different FXPAC phenotypes can involve different organs, such as brain in FXTAS or ovaries in FXPOI. Low levels of IGF2BP3 were detected in Human Protein Atlas (HPA) brain dataset[69], however we found that in iPSC-derived neurons used in our study, *IGF2BP3* expression was comparable to *FMR1*.

BET inhibitors were previously shown to decrease IGF2BP3 expression in cancer[46,47,50,70], however, the exact mechanism of action remains unclear. JQ1 impairs binding of bromodomain-containing protein 4 (BRD4) to chromatin in cancer cells[71], and ChIP-seq analysis revealed binding of BRD4 to *IGF2BP3* promoter region in PC3 cells[47]. This suggests a possible mechanism where JQ1 dissociates BRD4 from *IGF2BP3* promoter, thus leading to the decrease of *IGF2BP3* expression. Here, we found that administering JQ1, I-BET151, and isocorydine to cells expressing mutant *FMR1* mRNA diminished the protein levels of both IGF2BP3 and mutant FMR99xG but not the level of FMR16xG containing a short polyglycine tract. Although no link between bromodomain proteins and rCGGexp has been reported to date, the therapeutic potential of some BET inhibitors, including JQ1, was proposed previously for FXS[51] and ALS/FTD[72,73]. Epigenetic interactions between BET proteins and acetylated histones regulate gene transcription. A previous report has suggested that the histone acetylation state at the *FMR1* locus containing CGGexp is dynamic, and the use of histone acetyltransferase inhibitors lowers mutant *FMR1* mRNA in fibroblasts derived from PM carriers[74]. In this context, we cannot exclude that the administration of JQ1 and I-BET151 to cells expressing long FMRpolyG inhibits BET proteins, thus impairing epigenetic 'reading' of acetylated histones in treated cells, which may incur possible changes in the expression of many genes, leading to off-target effects. Nevertheless, we have shown that downregulation of IGF2BP3 level mediated by siRNA or BET inhibitors decreased cell death of FXTAS iPSC-derived neurons. Previous reports have shown that the expression of CGG RAN translation products reduces cell viability and causes neuronal death[17,18,52]. In current work, we confirmed these findings and described severe phenotypic changes in novel *C. elegans* model expressing fragment of FMR1 mRNA with expanded CGG repeats. Additionally, decreasing IGF2BP3 in both neuronal and animal models rescued FXPAC-like phenotypes, suggesting that IGF2BP3 can be a potential therapeutic target in FXTAS. This is an important finding, as current therapeutic approaches for FXTAS based on

antisense oligonucleotides[75] or small molecules[60,76,77] are still under development.

Our discovery that IGF2BP3 binds directly to the *FMR1*-5′UTR in vitro and regulates its translation prompted us to investigate the mechanism of this interaction. Previously, the RNA binding specificity of IGF2BPs was investigated using structural approaches, which revealed the requirement of the 'GG' motif for the binding of all IGF2BP family members[38,40]. Specific target RNA sequences, GGC-core elements in combination with the CA motif, are recognized by concerted interaction with all IGF2BP3 RNA binding domains[38]. In line with this requirement, the *FMR1* 5′UTR sequence contains multiple GGC elements with at least three CA motifs situated within the required distance of 22–25 nucleotides. Importantly, publicly available eCLIP data revealed IGF2BP3 reads coverage in the 5′UTR of *FMR1*[37]. Here, we found that a point mutation in one of these CA motifs, located upstream of the CGG repeats, completely abolished the regulation of FMRpolyG RNA and protein levels by IGF2BP3, demonstrating that the sequence context of the *FMR1* 5′UTR is critical for IGF2BP3 action in regulating RAN translation. Nevertheless, we cannot exclude the possibility that other CA motifs present upstream of CGG repeats can also mediate IGF2BP3 binding to *FMR1* transcript. Additionally, the stoichiometry of this complex requires further confirmation, as our in vitro translation data is limited to low range of protein concentrations that were tested.

CGG-RAN translation can be modulated via distinctive mechanisms, it was shown that its efficiency is dependent on the ribosomal composition, particularly the presence of RPS25 and RPS26, which translate particular set of mRNAs[36]. During translation elongation, ribosomal quality control proteins (NEMF, LTN1, and ANKZF1) respond to ribosomal stalling and degrade improperly synthesized peptides, while their depletion enhances CGGexp RAN translation and leads to accumulation of truncated peptides synthesized from expanded CGG repeats[31]. Importantly, proteasome subunit beta-5 (PSMB5) can impact both RAN translation and CGG toxicity mechanisms in FXTAS, but its excessive decrease shuts down global translation[60]. Another intriguing mechanism is the impairment of nuclear export of *FMR1* mRNA containing CGGexp, resulting in lower levels of RAN-translated proteins[43]. Our results suggest a model where IGF2BP3 binds to the CA motif and GGC sequence located in the 5′UTR of *FMR1* mRNA and causes a steric block for PIC scanning, favoring initiation at non-AUG codons. This action is even more pronounced if IGF2BP3 binds to a mutant 5′UTR of *FMR1*, where CGGexp forms very stable secondary structures and PIC scanning is already impeded. In the presence of IGF2BP3, the 43S PIC kinetics can be further slowed down, resulting in increased non-canonical biosynthesis of toxic FMRpolyG (Fig. 8). However, this hypothetical mechanism needs to be validated in further studies, including investigation of the effect of IGF2BP3 binding on FMR1 5′UTR RNA structure, ribosomal recruitment and translation initiation. Another parameter that should be considered is the potential impact of the localization of AGG interruptions in CGG repeats and the involvement of other proteins interacting with IGF2BPs in regulation of RAN translation.

In summary, we provide the evidence that IGF2BP3 directly binds to specific sequence in *FMR1* 5′UTR mRNA and positively regulates CGG-RAN translation. Moreover, two other members of the IGF2BP family also regulate the level of toxic FMRpolyG in the same direction as IGF2BP3. Importantly, the toxic effect of rCGGexp expression is significantly reduced upon IGF2BP3 insufficiency in FXTAS patients-derived neurons and in 99xCGG *C. elegans* model expressing mutant *FMR1* mRNA fragment. This discovery opens up new therapeutic possibilities, as the tested small molecule inhibitors of IGF2BP3 efficiently decreased FMRpolyG levels, which is a driver of FXTAS pathogenesis. Finally, presented here 99xCGG *C. elegans* model can now be used to investigate the molecular basis of FXPAC and to test novel therapeutic approaches.

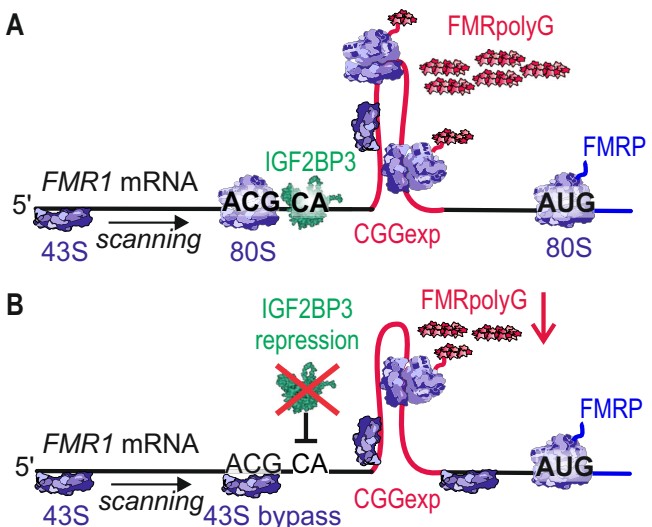

**Fig. 8 | A model of repeat-associated non-AUG initiated (RAN)-translation in the presence or absence of IGF2BP3. A** Mutant *FMR1* mRNA is translated to canonical FMRP protein, but expanded CGG repeats (CGGexp) located in its 5′UTR can also trigger RAN translation initiation at near-cognate start codons (ACG codon leading to the production of FMRpolyG is shown). IGF2BP3 binds to the RNA region containing CA motif located upstream of the CGG repeats and multiple GGC sequences located within the CGG repeats, and may cause a steric block for 43S preinitiation complex (PIC) scanning, thereby increasing initiation at near-cognate codons and biosynthesis of toxic FMRpolyG. **B** When IGF2BP3 is suppressed, either by knockdown or small molecule inhibitors, PIC scanning is facilitated and biosynthesis from non-cognate codons is initiated at a lower level, resulting in decreased FMRpolyG production.

## Methods

### Genetic constructs

The genetic construct *FMR1-99CGG* (Addgene plasmid #63091), was a kind gift from N. Charlet-Berguerand. This construct contains 5′UTR sequence of *FMR1* gene with ~99CGG repeats and located downstream GFP sequence[17] and was used to produce FMRpolyG. The *FMR1-16CGG* construct, was obtained by spontaneous contraction of CGG repeats from *FMR1-99CGG* template during bacterial growth. The *FMR1-delCGGs* construct, lacking CGG repeats, was generated from *FMR1-99CGG* template via inverse polymerase chain reaction (PCR), using InFusion Cloning technology (Takara), with primers F1/R1, followed by DNA fragments assembly with NEBuilder (NewEngland Biolabs). The already described *ATG-99CGG* construct[78], contains canonical ATG start codon instead of near-cognate ACG upstream of the repeats. Note that the part of *FMR1* 5′UTR upstream the start codon is lacking in this plasmid. Construct *ACG-16CGG-NLuc*, encoding 5′UTR sequence of *FMR1* gene with 16CGG repeats in frame with nanoluciferase and FLAG, was obtained via amplification of FMR1 5′UTR sequence from *ACG-16xCGG* Complete STOP plasmid[78] by primers F2/R2. The FLAG tag was added downstream nanoluciferase in *pNL1.1 CMV* construct (Promega) via inverse PCR with primers F3/R3. To ensure the transcription start site at the beginning of FMR1 5′UTR, the sequence of 109 nt between the CMV promoter and the FMR1 sequence was removed. The generated *pNL1.1 CMV-FLAG* backbone was linearized by primers F4/R4, mixed with FMR1 5′UTR amplicon and cloned via NEBuilder system. Construct *ACG-16CGG-mutCA-NLuc*, with mutation of CA nucleotides to TG in region predicted for binding IGF2BP3 was obtained by inverse PCR based on InFusion cloning, using primers F5/R5. GC-rich PCR product, corresponding to TMEM170 mRNA and serving as a control for RNA pull-down, was amplified by PCR from HEK293T cDNA, using primers F6/R6 and Phusion High-Fidelity DNA Polymerase (Thermo-Fisher). PCR product was used as a template in second PCR reaction with primers F7/R6, in order to add the T7 promoter sequence,

recognized by T7 RNA polymerase. Sequences of all constructs were confirmed by Sanger sequencing. All plasmids containing CGG repeats were transformed and isolated from NEB stable competent E.coli (New England Biolabs). *pDESTmycIGF2BP3* construct[79] used for the overexpression of IGF2BP3 was a gift from Thomas Tuschl (Addgene plasmid # 19879). *pMSCV PIG IMP-1-short* construct[80], used for overexpression of IGF2BP1, was a gift from David Bartel (Addgene plasmid # 21659). *pcDNA3-GFP-IMP2-2*, used for overexpression of IGF2BP2[81], was a gift from Alexandra Kiemer (Addgene plasmid # 42175). The list of all primers used in the study is available in a Supplementary Table 1.

### In vitro transcription

Biotinylated RNA probes were prepared by in vitro transcription on the *FMR1-99CGG* and *FMR1-delCGGs* constructs containing T7 polymerase promoter upstream to 5′UTR FMR1 sequence, and linearized by Avr II enzyme (New England Biolabs). Template for control GC rich RNA was obtained by PCR amplification of *TMEM107* gene fragment from HEK293T cDNA and adding T7 polymerase promoter by PCR. One μg of template DNA was used per 10 μL of transcription reaction containing 1x transcription buffer (Promega), 4 U of T7 polymerase (Promega), 10 mM DTT, 10 U of RNAsin (Promega), rNTPs (0.5 mM each), rCTP was mixed with Biotin-14-CTP analog (Invitrogen) in 10:1 ratio, to incorporate biotinylated cytidines in a random manner. Reaction was performed at 37 °C for 2 h. Transcribed RNA was precipitated using sodium acetate and resuspended in DNAse and RNase free water. All in vitro transcription reaction products were analyzed on 1% agarose gels with ethidium bromide (0.5 μg/ml) run in 1 x Tris-Borate-EDTA buffer at 70 V for 2 h.

For electromobility shift assay, RNA fragments were prepared in transcription reaction in 50 μL containing DNA template (~1 μg), 4 mM of each ribonucleotide (Promega), 1 x transcription buffer (Promega), 10 U of T7 polymerase (Promega), 10 mM DTT, 10 U of RNAsin (Promega). Reaction was performed at 37 °C for 2 h. Transcripts were submitted to DNAse treatment (Invitrogen), incubated with 1 U of Alkaline Phosphatase at 37 °C for 10 min, purified and concentrated on spin columns (A&A Biotechnology).

Template RNA for in vitro translation was obtained using mMES-SAGE mMACHINE T7 Transcription Kit (Invitrogen). *FMR1-16CGG* construct was linearized using PmlI enzyme (New England Biolabs). The reaction was assembled at room temperature according to the manufacturer's instructions and performed at 37 °C for 1.5 h. The transcribed RNA was treated with Turbo DNase (Invitrogen), purified using Clean-Up RNA Concentrator (A&A Biotechnology) and denatured (65 °C for 3 minutes) immediately before in vitro translation.

### Biotinylated RNA-protein pull down

Cellular extracts were prepared by lysing $2.5 \times 10^6$ SH-SY5Y or HEK293T cells in 200 μL of lysis buffer containing 50 mM Tris-Cl pH 7.5, 150 mM NaCl, 1% Triton X-100, 0.1% Na-deoxycholate supplemented with Halt Protease Inhibitor Cocktail (Thermo Fisher Scientific) for 30 min on ice. Lysates were cleared by centrifugation and 200 μL of supernatant was incubated for 20 min at 21 °C with 5 μg of RNA in 200 μL of 2xTENT buffer (100 mM Tris pH 7.8, 2 mM EDTA, 500 mM NaCl, 0.1% Tween) supplemented with RNAsin (Promega). RNA-protein complexes were then incubated with MyOne Streptavidin T1 DynaBeads (Thermo Fisher Scientific) for 20 min, followed by washing steps in 1x TENT buffer. For SDS-PAGE and Western blot analysis, bound proteins were released by heat denaturation in Bolt LDS Sample Buffer (Thermo Fisher Scientific) at 95 °C for 10 min. For MS analysis, bound proteins were released by incubation in digestion buffer containing 100 mM Tris HCl, 6 M Urea, 2 M Thiourea, 2% ASB-14, at 37 °C for 10 min.

### SDS-PAGE and Western blot (WB)

Harvested $0.24 \times 10^6$ HEK293T or HeLa cells or $0.5 \times 10^6$ SH-SY5Y cells were lysed in lysis buffer (50 mM Tris-Cl pH 7.5, 150 mM NaCl, 1%

Triton X-100, 0.1% Na-deoxycholate) supplemented with Halt Protease Inhibitor Cocktail (Thermo Fisher Scientific) for 30 min on ice, followed by sonication and centrifugation at 14,000 g for 10 min at 4 °C. Extracted proteins were denatured in Bolt LDS Sample Buffer mixed with Bolt Reducing agent (Thermo Fisher Scientific) at 95 °C for 5 min. Proteins were separated in Mini Gel Tank (Invitrogen) using Bolt™ 4–12% Bis-Tris Plus gels (Thermo Fisher Scientific) in Bolt™ MES SDS Running Buffer (Thermo Fisher Scientific). For silver staining analysis, gels were stained with Silver Stain for Mass Spectrometry kit (Pierce) according to the manufacturer's instructions. For WB, proteins were electroblotted to PVDF membrane (0.2 μm, GE Healthcare) for 1 h at 100 V in ice-cold Bolt™ Transfer Buffer (Thermo Fisher Scientific). Membranes were blocked in room temperature for 1 h in 5% skim milk (Sigma) in Tris-buffered saline with 0.1% Tween-20 (TBS-T). Incubation with antibodies was performed in following conditions: rabbit anti-FMRP antibody (ab17722, Abcam) 1:1000, rabbit anti-IGF2BP3 antibody (ab177477, Abcam) 1:1000, anti-FMRpolyG antibody (MABN1788, Sigma) 1:1000, anti-Vinculin (sc-73614 HRP, SantaCruz Biotechnology) 1:4000, anti-GAPDH antibody (sc-47724 HRP, SantaCruz Biotechnology) 1:2000, diluted in 5% skim milk in TBS-T for 16 h at 4 °C. Membranes were washed in TBS-T and incubated with horseradish peroxidase-conjugated secondary antibodies, anti-rabbit (AS09 602, Agrisera) 1:4000 in TBS-T or anti-mouse (A9044, Sigma) 1:10,000 in TBS-T for 1 h at RT. Membranes were washed in TBS and covered with Immobilon Forte Western HRP Substrate (Sigma) according to the manufacturer's instructions. Images were captured with the use of G:Box Chemi-XR5 (Syngene) and ChemiDoc Imaging System (BioRad) and quantified using GeneTools 4.02 (Syngene) or ImageLab (BioRad), respectively. Relative protein level was normalized to Vinculin.

### Proteomic digestion

Proteins that bound to biotinylated RNA probes (FMR1-99CGG RNA, FMR1-delCGG RNA and GC-rich RNA as a control) were eluted in digestion buffer as described above. Each dataset included three independent pull-downs ($N = 3$). Then, samples were reduced by the addition of 3 μL of 5 M dithioerythritol (Sigma) for 1 h at RT, and alkylated with 6 μL of 5 M iodiacetamide (Sigma) for 30 min at RT in dark. After this step, 1 μg of Trypsin/Lys-C (Promega) solution was added, followed by incubation for 3 h at 37 °C. Then, sterile MilliQ water was added to the samples and digestion was continued at 37 °C for 16 h. After digestion, samples were desalted using solid phase extraction (SPE). 180 μL of 0.2% trifluoroacetic acid (Sigma) was added into every sample prior to the SPE. Samples were purified using C18 Isolute columns (Biotage) according to the manufacturer's instructions. Collected peptides were dried in speed-vac and submitted to proteomic analysis.

### Mass spectrometry (MS) analysis

LC-MS analysis was performed in Mass Spectrometry Laboratory, Institute of Biochemistry and Biophysics, Polish Academy of Sciences, Pawińskiego 5a Street, 02-106 Warsaw, Poland. The samples were analyzed using a nanoAcquity UPLC system (Waters) connected to a QExactive mass spectrometer (Thermo Scientific). Peptides were trapped on a C18 precolumn (180 μm x 20 mm, Waters) with 0.1% formic acid (FA) in water as the mobile phase. Then, transferred to a nanoAcquity BEH C18 column (75 μm x 250 mm, 1.7 μm, Waters) using an acetonitrile (ACN) gradient (0–35% ACN over 160 min) with 0.1% FA at a flow rate of 250 nL/min. Data was acquired in data-dependent mode, selecting the top 12 precursors for MS2 within an isolation window of 3 m/z. Full MS scans were conducted for 300–2000 m/z at a resolution of 70,000, with a maximum injection time of 60 ms and an AGC target value of 1e$^6$. MS2 scans were performed at a resolution of 17500, with a maximum injection time of 300 ms and an AGC target of 2e$^5$. Dynamic exclusion was applied for 30 s.

### MS data analysis

Raw data obtained from the LC-MS/MS runs were analyzed using MaxQuant v2.6.4.0 using the label-free quantification (LFQ) with default parameters. UniProtKB database for reviewed human canonical and isoform proteins of May 2023 was used. The false discovery rate (FDR) at the peptide spectrum matches and protein level was set to 0.01; variable peptide modifications: oxidation (M) and acetyl (N-term), fixed modification: carbamidomethyl (C), two missed cleavages were allowed. Statistical analyses were performed using Perseus software v2.1.2.0 after filtering for "reverse", "contaminant" and "only identified by site" proteins. The LFQ intensity was logarithmized (log2[x]), and imputation of missing values was performed with a normal distribution (width = 0.3; shift = 1.8). Proteomes were compared using t-test statistics with a permutation-based FDR of 5% and P-values < 0.05 were considered to be statistically significant.

### Gene Ontology (GO) analysis

Lists of proteins that were identified by MS as binding to FMR1-99CGG, FMR1-delCGGs, and GC-rich RNAs were submitted to gene ontology (GO) analysis using Panther[82] software, released 2024-01-17. Analysis type was Statistical Overrepresentation test, with annotation sets: GO biological process complete, GO molecular function complete, and GO cellular component complete. Used test type was Fisher's exact with the use of Bonferroni correction for multiple testing.

### Electrophoretic Mobility Shift Assay (EMSA)

RNA fragments were prepared using in vitro transcription, as described above. For 5′-end radiolabeling, 2 pmol of RNA was incubated with 2 pmol of [γ-P32] ATP, 1 U of RNasin (Promega), and 10 U of T4 polynucleotide kinase (Thermo Scientific) in 1x reaction buffer in a total volume of 10 μL and at 37 °C for 30 min. The excess of [γ-P32] ATP was removed on spin columns (A&A Biotechnology). EMSA was carried out by incubating 5′ radiolabelled RNA with recombinant IGF2BP3 protein (Origene) of indicated concentrations (ranging from 0 to 100 nM) in a volume of 10 μL, in buffer containing 10 mM Tris-Cl pH 7.5, 50 mM KCl, 1 mM EDTA, 0.05% Triton X, 5% Glycerol and 1 mM DTT, and incubated at 37 °C for 20 min. The samples were run on a native 6% polyacrylamide gel in 0.5x TBE at 100 V for 1 h. The gel was subsequently dried, and the signal was detected overnight, visualized on phosphoimager Amersham Typhoon IP (GE Healthcare), and quantified using Multi Gauge 3.0 software (FujiFilm).

### Filter Binding Assay (FBA)

FBA was carried out by incubating 5′-end radiolabelled RNA with recombinant IGF2BP3 protein (Origene) of indicated concentrations (ranging from 0 to 100 nM) in a volume of 10 μL, in buffer A containing 10 mM Tris-Cl pH 7.5, 50 mM KCl, 1 mM EDTA, 0.05% Triton X, 5% Glycerol and 1 mM DTT, and incubated at 37 °C for 20 min. Next, samples were loaded under vacuum-induced pressure onto two membranes: nitrocellulose (Protran BA 85, Whatman) and nylon (Hybond™ N + , Amersham) placed in dot blotter apparatus. Following wash with buffer A, RNA- protein complexes remained on the top nitrocellulose membrane, while free RNA was retained on a nylon membrane. The signal from membranes was detected O/N, visualized on phosphoimager Amersham Typhoon IP (GE Healthcare) and quantified using Multi Gauge software (FujiFilm). The dissociation constant (Kd) of the RNA/IGF2BP3 interaction was calculated in the GraphPad program using the following equation: one site specific binding curve (Y= Bmax*X/(Kd + X)).

### Cell culture and transfection

The human HEK293T, HeLa and SH-SY5Y cells were grown in a high glucose DMEM medium with L-Glutamine (Thermo Fisher Scientific) supplemented with 10% fetal bovine serum (FBS; Thermo Fisher Scientific) and 1x antibiotic/antimycotic solution (Sigma). S-95xCGG and

S-16xCGG cells were grown in DMEM medium containing certified tetracycline-free FBS (Biowest). All cells were grown at 37 °C in a humidified incubator containing 5% $CO_2$. Cell lines were routinely tested for mycoplasma using mycostrip test (Invivogen) and found negative.

siRNA were delivered by reverse transfection protocol using jet-PRIME® reagent (Polyplus) according to the manufacturer's instructions. Plasmids were delivered 24 h post siRNA silencing using jetPRIME® reagent (Polyplus) and manufacturer's transfection protocol and harvested 48 h post siRNA silencing and 24 h post transient plasmids expression. The list of all siRNAs and appropriate concentrations used in the study is available in a Supplementary Table 2.

### RNA isolation and quantification of mRNA level
Cells were harvested in TRI Reagent (Thermo Fisher Scientific), and total RNA was isolated with Total RNA Zol-Out D (A&A Biotechnology) kit according to the manufacturer's protocol. 500 ng of RNA was reversely transcribed using High Capacity Reverse Transcription Kit with RNAse inhibitors and random primers (Invitrogen). Quantitative real-time RT-PCRs were performed in a QuantStudio 6 Flex System (Thermo Fisher Scientific) using Maxima SYBR Green/ROX qPCR Master Mix (Thermo Fisher Scientific) with 5 ng of cDNA in each reaction. Transgene FMR1-99CGG mRNAs were amplified with primers F8/R8 with a note that reverse primer was anchored in GFP sequence in order to distinguish endogenously expressed *FMR1* transcripts (amplified with primer F9/R9 pair). Reactions were run at 58 °C annealing temperature and Ct values were normalized to *GAPDH* mRNA level (amplified with primer F10/R10 pair). Fold differences in expression level were calculated according to the $2^{-\Delta\Delta Ct}$ method.

### Flow cytometry
Cells were harvested 72 h post siRNA delivery and 48 h post transient plasmids expression by trypsinization followed by centrifugation. Cell pellets were suspended in 200 μL of PBS. Dead cells were stained with propidium iodide (PI) at final concentration 1 μg/μL and excluded from analysis. Cells were analyzed with Sony MA900 cell sorter. GFP was excited by 488 nm laser and collected after a 525/50 nm bandpass (BP) filter. For each sample, 10.000 GFP-positive cells were collected. Threshold for GFP-positive cells was set based on signal from non-transfected cells. Forward scatter high (FSC-H) and forward scatter width (FSC-W) were plotted to permit doublet discrimination.

### Microscopic analysis of FMRpolyG-GFP aggregates
FMRpolyG aggregates were detected by fluorescence microscopy as described previously[78]. Briefly, HeLa cells were grown on 48-well plates, siRNA and plasmid transfections were delivered as described in Cell culture and transfection section. Fourty-eight hours post siRNA delivery and 24 h post FMR1-99CGG delivery, HeLa cells were incubated in standard growth medium with final concentration of 5 μg/ml of Hoechst 33342 (Thermo Fisher Scientific) for 10 min. Images were taken with Axio Observer.Z1 inverted microscope equipped with A-Plan 10×/ 0.25 Ph1 or LD Plan-Neofluar 20×/0.4 Ph2 objective (Zeiss), Zeiss Colibri 7 excitation band 385/30 nm, emission filter 425/30 nm (Hoechst) and Zeiss Colibri 7 excitation band 469/38 nm, emission filter 514/30 nm (GFP), Zeiss AxioCam 506 camera and ZEN 2.6 pro software. Presented values were quantified from 10 images, number of cells and aggregates were calculated using ImageJ and AggreCount plugin.

### Cytoplasm/nucleus fractionation
For fractionation of the nuclei and cytoplasm, HEK293T cells were seeded on T25 flasks, and transfected with siRNA and FMR1-99CGG plasmid as described above. Cells were harvested 24 h post plasmid delivery and fractionated using PARIS™ Kit (Invitrogen), according to manufacturer's instructions. Total, cytoplasmic and nuclear RNA was isolated using Filter Cartridges supplied in the PARIS™ Kit. DNA

contamination was removed using TURBO DNA-free™ kit (Invitrogen) according to manufacturer's instructions. cDNA synthesis and qPCR were performed as described in "RNA isolation and Quantification of mRNA" section.

### Apoptosis assay
HeLa cells were grown on 96-well plate and transfected with siIGF2BP3 or siControl at 50 nM concentration using JetPrime reagent (Polypus). 24 h post-siRNA transfection, plasmid encoding *FMR1-99CGG* was delivered to the cells using JetPrime reagent (Polypus). 46 h post siRNA and 22 h post plasmid delivery, apoptosis was tested with RealTime-Glo™ Annexin V Apoptosis Assay (Promega) according to the manufacturer's instructions. Luminescence signal corresponding to apoptosis was measured 4 h after addition of the reaction mix (50 h post siRNA and 26 post-plasmid delivery, respectively), using SPARK microplate reader (TECAN).

### In vitro translation
Flexi Rabbit Reticulocyte Lysate System (Promega) was used for in vitro translation. In vitro translation reactions contained 10 nM mRNA, 70% rabbit reticulocyte lysate, 10 μM amino acid mix minus leucine, 10 mM amino acid mix minus methionine, 70 mM KCl, 0.5 mM MgOAc, 0.8 U/μL RNasin (Promega) and recombinant IGF2BP3 protein (Origene) at various concentrations (0, 10, 50 nM). Reactions were performed at 30 °C for 60 min and terminated by incubation on ice. Reactions were then sonicated, diluted 1:10 in concentrated Bolt LDS Sample Buffer mixed with Bolt Reducing agent (Thermo Fisher Scientific) and incubated at 70 °C for 15 min to denature the proteins. The prepared samples were then analyzed by SDS-PAGE and WB.

### iPSC culture
Two human FXTAS iPSC lines, expressing 72 and 83 CGG repeats in the *FMR1* 5′UTR, were a kind gift from Nicolas Charlet-Berguerand, and were described earlier[17]. Legal ethical approval and informed consent from the patients were obtained to generate and reuse these iPS cells for scientific and research purpose. Two human control iPSC lines, harboring 28 and 29 CGG repeats in the 5′UTR of *FMR1* were reprogrammed in our laboratory using Episomal iPSC Reprogramming Vectors (Thermo Fisher Scientific). iPSCs were cultured in StemFlex (Thermo Fisher Scientific) medium on Geltrex-coated (Thermo Fisher Scientific) plates at 37 °C humidified incubator containing 5% $CO_2$. Cells were passaged every 3–4 days using PBS-EDTA. Cell lines were routinely tested for mycoplasma using mycostrip test (Invivogen) and found negative.

### iPSC neuronal differentiation
iPSCs were differentiated into neurons as described previously[17] with modifications. Briefly, 2 million iPSCs were plated on one Geltrex-coated well (Thermo Fisher Scientific) of 6-well plate in Essential 8 medium (Gibco) with 10 μM Y-27632 dihydrochloride (MedChemExpress). The next day, medium was changed to NFS medium (N2B27 supplemented with 20 ng/mL FGF2 (Peprotech), 0.25 μM LDN-193189 (Sigma) and 10 μM SB431542 (MedChemExpress)), and cells were grown for 10 days with daily medium change. After the appearance of neural rosette, cells were passaged onto poly-L-ornithine (Merck) and laminin (Merck)-coated 48-well plates, and NFS medium was replaced with NSC medium (N2B27 supplemented with 20 ng/mL hBDNF (Peprotech), 1 μM LY-411575 (MedChemExpress) and 0.2 μM Ascorbic Acid (Sigma)). Media change was performed every two days and cells were differentiated for 21 days in NSC medium. For measuring the impact of IGF2BP3 KD on *FMR1* level, control siRNA and siRNA targeting IGF2BP3 were delivered by lipofection using RNAiMAX (Thermo Fisher Scientific) after 14 days of differentiation and cells were harvested 60 h post siRNA treatment. For measuring the number

of FMRpolyG inclusions experiment, control siRNA and siRNA targeting IGF2BP3 were delivered by lipofection using RNAiMAX (Thermo Fisher Scientific) every 7 days, starting from day 7 after beginning of differentiation.

## Immunofluorescence

For immunofluorescence, iPSC-derived neurons were grown on poly-L-ornithine (Merck) and laminin (Merck) cell culture chamber slides (Ibidi). After 21 days of differentiation, cells were fixed in 4% paraformaldehyde/PBS for 15 min at room temperature, washed with PBS, and permeabilized in 0.5% Triton X-100/PBS and blocked by incubation for 45 min with 5% normal donkey serum (Jackson ImmunoResearch) 0.1% tween-20 in PBS. Samples were incubated overnight at 4 °C with primary antibody against FMRpolyG (Millipore 8FM, 1:100) and MAP2 (ab5392, Abcam, 1:5000). The next day, slides were washed with 0.1% Tween-20 in PBS before incubation with secondary antibodies conjugated with Alexa Fluor 488 or Alexa Fluor 647 for 1 h at room temperature, and the excess of secondary antibody was removed by two washes in 0.1% Tween-20 in PBS. The samples were then incubated with Hoechst 33342 (Thermo Fisher Scientific) as a counterstain. Slides were examined using a confocal microscope Leica SP8 equipped with white laser, HyD S detectors, and HC PL APO CS2 40x/1.10 WI objective. Cells were imagined as z-stacks with voxel size 0.57 × 0.57 × 0.42 μm. Alexa 488 (FMRpolyG), DAPI, and Alexa 647 (neuronal marker MAP) were imagined as separate channels with fluorochrome-specific excitation and emission settings. All images utilized for aggregates counting were done with the same microscopy settings. FMRpolyG aggregates as high fluorescence intensity objects were identified in Fiji ImageJ by Auto-Threshold (method: Internodes) on Maximum Z-Projection images with Alexa 488 signal. For avoiding non-specific aggregates, only objects identified in 10 μm distance from DAPI-stained nucleus were counted. At least 350 cells per line were imagined for counting aggregates.

## Necrosis and apoptosis assay of FXTAS iPSC-derived neurons

Control and FXTAS iPSC-derived neurons were grown on 96-well plate covered with laminin for 10 days and transfected with siIGF2BP3 or siControl at 50 nM concentration using RNAiMAX (Thermo Fisher Scientific). 48 h post-siRNA transfection, apoptosis and necrosis were tested with RealTime-Glo™ Annexin V Apoptosis and Necrosis Assay (Promega) according to the manufacturer's instructions. Luminescence or fluorescence signal corresponding to apoptosis or necrosis, respectively, was measured at different time points starting from 1 h after addition of the reaction mix, using SPARK microplate reader (TECAN).

## C. elegans maintenance

Experiments conducted using *Caenorhabditis elegans* are not subjected to institutional ethical approval for animal research. Work was carried out in accordance with established guidelines for handling *C. elegans*. *C. elegans* strains were maintained according to standard methods[83]. Animals were grown on nematode growth medium (NGM) agar plates seeded with *Escherichia coli* OP50 as a food source. Healthy populations were maintained by transferring worms to fresh OP50-seeded plates every 3–4 days to prevent starvation and overcrowding. To synchronize the worms, we extracted the eggs from gravid adults with bleaching solution (30% (v/v) Sodium hypochlorite solution 5% pure Warchem Catalog number: 7681-52-9), 0.75 M KOH). The eggs were left in M9 buffer (42 mM Na$_2$HPO$_4$, 22 mM KH$_2$PO$_4$, 86 mM NaCl, 1 mM MgSO$_4$) without any food for 12–16 h, thus generating synchronised, arrested L1 stage worms, which were then plated on plates with bacterial food as described above.

## Transgenic animal generation

Transgenic *C. elegans* animals were generated by microinjection of DNA into the gonads of young adult Bristol N2 worms, following standard procedures[84]. Injection mixtures contained 100 ng/μL of the plasmid containing FMRpolyG ORF and 20 ng/μL of a co-injection marker (pCFJ104 P*myo-3::mCherry*). Injections were performed using a Narishige micromanipulator and an Eppendorf InjectMan 4 micro-injection system.

Injected worms (P0s) were singled onto NGM plates seeded with *E. coli* OP50 and incubated at 20 °C. F1 progeny were screened for the presence of the co-injection marker under a fluorescent dissecting microscope (Nikon SMZ800N). Animals expressing the marker were picked to establish stable transgenic lines carrying extrachromosomal arrays.

## Brood size counting

Brood size counting was done according to the protocol described in Gudipati et al.[85]. *C. elegans* hermaphrodites were singled onto 55 mm plates seeded with *E. coli* OP50 at the fourth larval stage and incubated at 25 °C. These animals were referred to as P0s. Once egg-laying commenced, P0s were transferred to fresh plates, and this process was repeated until egg-laying had completely stopped. The number of F1 viable progeny was counted. The data was represented as a violin with each dot referring to the total number of viable progenies from a single P0.

## Thrashing assay

Locomotor activity of *C. elegans* was evaluated using a thrashing assay. L4 stage larvae were transferred onto fresh NGM plates seeded with *E. coli* OP50 and maintained at 25 °C for three days, to obtain synchronised 3-day-old adult animals. For thrashing analysis, individual worms were placed into 1 ml of M9 buffer on an empty 55 mm plate. After a 1 min acclimatization period, worms were observed under a dissecting microscope. A single thrash was defined as a complete bending movement in which the head, mid-body, and tail of the worm were oriented in opposite directions, forming a "C" shape. Thrashing events were manually counted for 1 min using a dissecting microscope (Leica S9 E). At least 15–20 animals were scored per condition for each biological replicate. Data was represented as a violin where each dot corresponds to the thrashes per minute of a single 3-day adult animal.

## Longevity assay

Age-synchronized L4 animals were transferred to fresh NGM plates seeded with OP50. These plates were labelled as Day 0 plates (the beginning of the experiment) and maintained at 25 °C. About 220–235 animals were scored for each genotype. The animals were transferred to fresh plates every other day until the end of their reproductive phase to prevent mixing up with the progeny. Animals were scored daily for survival by gently poking their heads with a platinum worm pick. If the animals did not respond to multiple poking, they were scored as dead and were removed from the plate. Animals missing from the plate or showing bagging and bursting phenotypes were scored as censored. Survival data were compiled and analysed in GraphPad Prism version 9 using the Kaplan–Meier method, and statistical significance was assessed with the log-rank test.

## C. elegans image acquisition

Animals were mounted on a 3% w/v agarose pad on a clean glass slide containing 10 μL of 10 mM levamisole solution. Images were acquired on a Nikon ECLIPSE NI-E 931349 using NIS Elements software. The Fluorescent and Differential Interference Contrast (DIC) images were obtained with a 40x oil immersion objective. Selected representative images were processed using the Fiji software.

Fluorescence lifetime images of FMRpolyG-GFP expressed in the head of 99xCGG model were obtained with Leica SP8 confocal microscope equipped with FAst Lifetime CONtrast (FALCON) module. HC PL APO CS2 40x/1.10 WI objective was used. GFP was excited with 488 nm and detected with HyD S detector in 494-525 nm range.

Lifetimes from pharynx were marked with ROI and calculated in Las X FLIM module by fitting with single exponential in reconvolution fit model. For FMRpolyG fluorescence measurements, 99xCGG and 99xCGG; *imph-1Δ* animals were imaged as z-stacks. The same settings of imaging were applied for all *C. elegans* genotypes. The EGFP intensity (RawIntDen; ImageJ) was measured in pharynx area of 26 animals for each strain, from Sum Intensity Projection images.

## Statistics and reproducibility

All data obtained in this study were processed and analyzed using Microsoft Excel and GraphPad Prism. Statistical analysis of all experiments performed on cell lines was done using unpaired two-tailed Student's t-test. Tests that result in $P < 0.05$ have been reported to be statistically significant. The symbols; *,**,***,**** represent values of $P < 0.05$, $P < 0.01$, $P < 0.001$, $P < 0.0001$ respectively. Error bars represent standard deviation (SD). All cellular and in vitro experiments presented in this work were repeated at least two times with similar results.

## Reporting summary

Further information on research design is available in the Nature Portfolio Reporting Summary linked to this article.

## Data availability

The raw mass spectrometry data sets have been deposited at the ProteomeXchange Consortium via the PRIDE partner repository with identifier PXD053905. The raw western blot images generated in this study are provided in the Supplementary Figs.; The list of identified proteins binding to the following RNA baits: FMR1-99CGG, FMR1-delCGGs, GC-rich, and 23CGG, generated in this study are provided in the Supplementary Data 1.

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

## Acknowledgements

This work was supported by the National Science Center (Poland) [2019/35/D/NZ2/02158 to A.B., 2020/38/A/NZ3/00498 to K.S., 2021/42/E/NZ1/00336 and 2022/45/B/NZ2/02183 to R.K.G.] and the European Union's Horizon 2020 Research and Innovation Program under the Marie Sklodowska-Curie grant agreement [No. 101003385 to A.B.]. Funding for open access charge: Initiative of Excellence–Research University at Adam Mickiewicz University, Poznan, Poland. Some *C. elegans* strains were provided by the Caenorhabditis Genetics Center (CGC), which is funded by NIH Office of Research Infrastructure Programs (P40 OD010440). We thank Nicolas Charlet-Berguerand for sharing FXTAS iPSC lines. We thank Katarzyna Taylor for help with EMSA assays.

## Author contributions

A.B., R.K.G., and K.S. conceptualized the study and acquired funding. A.B., D.S., T.S., I.B., and D.N. performed the experiments and analyzed the data. W.J.Sz. and M.B. provided the equipment and intellectual input on iPSC-related experiments. A.B. wrote the manuscript with editing contributions from K.S. All authors reviewed the manuscript.

## Competing interests

The authors declare no competing interests.
