## [Transparent Peer Review file · Nature Communications]

IGF2BPs directly regulate the noncanonical translation of toxic proteins from mutant FMR1 mRNA containing expanded CGG repeats

Corresponding Author: Professor Krzysztof SOB CZAK

A version of this paper was originally rejected for publication by Nature Communications, however that decision was reconsidered after appeal by the authors.

Version 0:

Reviewer comments:

Reviewer #1

(Remarks to the Author)

Fragile X Tremor and Ataxia Syndrome (FXTAS) is a rare neurodegenerative disease caused by an expansion of CGG repeats located within the 5'UTR of the FMR1 gene. These CGG repeats are translated into a novel protein, FMRpolyG, which is prone to form cellular inclusions, and which expression is toxic in cell and animal models. Importantly, translation of these repeats depends on initiation at near-cognate start codons (GUG, ACG) located upstream of the repeats. Fidelity of translation initiation at near cognate codons vs canonical AUG start sites is closely regulated by several proteins (eIF1, eIF 2 and eIF5, among others), and thus of potential clinical interest in FXTAS to modulate expression of the toxic FMRpolyG protein.

In that aspect, Baud and collaborators found that the IGF2BP RNA binding proteins binds to the FMR1 5'UTR sequence and regulate FMRpolyG expression. This is an important finding as it open routes toward identifying a therapeutic approach for this devastating syndrome. Overall, presented data are solid and clear, experiments are technically well controlled, and the results are novel and of general interest for people working on FXTAS and/or on regulation of translation initiation. However, interest for this work is tempered by several points:

Main comments:

- All data have been generated in vitro or using transformed cell lines (HeLa, HEK or SH-SY-5Y), thus this work, especially the use of drugs inhibiting IGF2BP as a potential therapeutic approach, suffers from a lack of validation in physiological iPS FXTAS cell or animal models.

- IGF2BP is a family of 3 proteins, hence effects of overexpressing or siRNA-mediated depletion of IGF2BP3 in Figure 2 could be fused to the analysis of IGF2BP1 & 2 in Figure 4. Importantly, siRNA-depletion needs to consider compensation between IGF2BP proteins, and a depletion of 2 or all 3 members by siRNA should be investigated. This is especially important as according to database, RNA expression of these proteins change according to cell type: only IGF2BP3 mRNA is expressed in SH-SY-5Y, but both IGF2BP1 and 3 transcripts are expressed in HeLa and all three are predicted to be expressed in HEK293, etc.). Thus, levels of these 3 proteins in the cell lines analyzed by the authors (with and without their corresponding siRNA treatment) should be presented. Moreover, out of the 3 IGF2BP proteins, only IGF2BP2 mRNA is predicted to be expressed in human brain, and only weakly (~10 nPTM in human cerebellum), thus this could be discussed for a future potential therapeutic strategy. Alternatively, as mRNA and protein levels not always correlate, a western on various human or mouse tissue, including the ones affected in FXTAS, showing expression of these protein would strengthen a proposed strategy based on decreasing IGF2BP proteins levels.

- Minor comments: the RNA structure presented in figure 3C would gain in clarity with symbols or colors indicating FMRpolyG and its near-cognate codons different from the color or symbol of other proteins/RAN product and their start sites. Same comment for the CA element required for IGF2BP binding, etc.

- Finally, various recent reports (including by the same Sobczak's group) show modulation of CGG repeat translation at the initiation, elongation or degradation level, notably by PSMB5 (Kong et al., 2022), RPS25/26 (Tutak et al., 2024), ANKZF1 (Tseng et al., 2024), etc. Thus, it could be of interest to show extend of the IGF2BP effect compared to other reported modulators.

Reviewer #2

(Remarks to the Author)

The manuscript by Sobczak and colleagues investigates the role of insulin-like growth factor 2 mRNA-binding protein 3 (IGF2BP3) in the regulation of noncanonical translation of toxic proteins associated with CGG repeat expansions in the FMR1 gene, a mechanism implicated in fragile X-associated tremor/ataxia syndrome (FXTAS). The study demonstrates that IGF2BP3 binds to the 5' untranslated region (UTR) of FMR1 mRNA, specifically near expanded CGG repeats, and enhances repeat-associated non-AUG (RAN) translation, which produces neurotoxic polyglycine proteins (FMRpolyG). Below are some of my specific comments:

1. In Vivo Validation of IGFBP3 Binding to CGG Repeats: The authors present comprehensive in vitro data showing IGFBP3 binding to CGG repeats; however, without in vivo confirmation of this interaction, the physiological relevance remains uncertain. Demonstrating IGFBP3 binding to CGG repeats within the context of living cells or in a relevant animal model would significantly strengthen their claims. Specifically, assays such as RNA immunoprecipitation (RIP) coupled with qPCR or cross-linking and immunoprecipitation (CLIP) would validate IGFBP3's interaction with endogenous FMR1 mRNA. This data is essential to rule out potential artifacts from in vitro binding conditions, which may not fully mimic the cellular environment. The absence of in vivo validation leaves a critical gap, questioning the translatability of these findings to biological settings.

2. Limitations of Overexpression Systems in CGG Translation Assays: The reliance on overexpression systems to study CGG repeat translation limits the applicability of the findings. Overexpression can alter cellular dynamics, potentially leading to non-physiological binding events or artifacts that would not occur at endogenous levels. Thus, it would be critical for the authors to assess RAN translation using native FMR1 transcript levels. Including in vivo or endogenously expressed FMR1 models would validate the observed regulatory effects and reinforce the physiological relevance of IGFBP3's role in CGG translation. Without this, it is challenging to determine if the reported effects are an artifact of overexpression.

3. Specificity of Small Molecule Inhibitors on IGF2BP Paralogs: While the authors indicate that small molecule inhibitors reduce IGFBP3 expression, it remains unclear whether these inhibitors selectively target IGFBP3 or also affect IGF2BP1 and IGF2BP2. Given the close structural similarity among IGF2BP family members, cross-reactivity is likely, and off-target effects could complicate the interpretation of results. Additional experiments are needed to dissect whether these inhibitors selectively target IGFBP3 without interfering with other paralogs. For example, parallel knockdown or CRISPR knockout experiments of IGF2BP1 and IGF2BP2 would clarify if these paralogs contribute similarly to CGG repeat translation. This analysis needs to be more robust to the therapeutic implications of their findings, as off-target effects may reduce the efficacy or safety of IGFBP3 inhibitors in clinical applications.

4. Mechanistic Insight into IGFBP3's Role in RAN Translation: The manuscript presents IGFBP3 as a regulator of RAN translation, yet it does not sufficiently explore the mechanistic basis underlying this regulation. Specifically, further investigation into whether IGFBP3 impacts translation initiation factors, ribosomal recruitment, or RNA secondary structures would provide greater mechanistic clarity. This would also address whether IGFBP3's influence is generalizable across other noncanonical translation events or specific to CGG repeats. Such insights are crucial for understanding the broader implications of IGFBP3 function in RNA biology and neurodegenerative disorders.

5. Experimental Controls and Replicability: There needs to be more discussion regarding experimental controls and biological replicates, particularly for complex assays like in vitro translation and protein-RNA interaction studies. Including appropriate controls—such as non-targeted RNA for pull-down assays and non-binding RNA mutants—would strengthen the reliability of the data. Additionally, clear reporting of the number of biological replicates would enhance reproducibility and lend robustness to the findings.

Reviewer #3

(Remarks to the Author)

The study explores the role of IGF2BP3, in aiding the translation of FMR1 mRNA with expanded CGG repeats. Using RNA-pull down followed by mass spectrometry and western blots, the authors identified IGF2BP3 binds to FMR1 mRNA probs either with or without CGG repeats. IGF2BP3 knockdown decreased the protein level of FMRpolyG but increased mRNA level, while OE of IGF2BP3 increased the protein level of FMR plyG but no changes on mRNA level. Then, by replacing the near-cognate ACG codon with AUG in FMRplyG mRNA as a mutant FMR16xG, OE or KD of IGF2BP2 does not affect either protein or mRNA level of FMR16xG. Next, the authors showed that other IGF2BP family proteins including IGF2BP2 and IGF2BP1 also have similar effects in aiding FMRpolyG biosynthesis. Targeting IGF2BP3 by some BET inhibitors showed decreased protein levels of IGF2BP3 as well as FMR99xG. The whole part of the work stands overall on less solid ground, with unclear rationale for looking at sometimes FMR16xG but sometimes FMR99xG expression levels and with insufficient controls to determine that this is indeed the functional ability of IGF2BP3 to modulate FMRpolyG expression. In general, no experiment in the study is up to current technological standards, and no robust conclusion can be drawn in the absence of

unbiased readouts, including at the very least measuring the levels of endogenous FMR1, FMR16xG, FMR96xG simultaneously upon KD or OE of IGF2BP2 in the key experiments, as well as the rescue experiments to confirm FMRpolyG is the essential target of IGF2BP2. Additional comments below, in no particular order of importance.

In figure 1, the authors concluded that IGF2BP3 binds to the FMR1 5'UTR in a CGG-independent manner. To be convinced, biotinylated RNA probes designed with expanded CGG repeats but without FMR1 5' UTR region should be used as a negative control instead of GC-rich RNA in figure 1A, as well as for all the protein binding assay experiments in figure 1B-D.

Figure1D, the authors turned to use FMR1-5'utr RNA containing 16CGG repeats in filter binding assay. Why not use FMR1-99CGG and FMR1-delCGG in this binding assay and do a comparison? What is the rationale to use FMR1-16CGG as research objective suddenly? In addition, only showing one Kd value means nothing. Please show the Kd value with either one negative or positive control.

Figure 2A, if as authors concluded that IGF2BP3 binds to FMRP in a CGG-independent manner, why is there no influence on normal FMRP protein level? how was the mRNA level of endogenous FMRP? The authors need to provide additional data to clarify this point.

Figure 2D-F, the study should also perform with FMR16xG. Additionally, the apoptosis induced by FMR99xG upon KD of IGF2BP3 is not convinced unless rescue experiments are performed.

RIP-seq should be performed to find the exact binding sites in FMRpolyG mRNA. Otherwise, can authors check the RIP-seq public data to find the motif?

AUG-FMR16xG, normal FMRP should also be tested in figure 3A as controls.

In fig 3B, more IGF2BP3 concentrations should be added to confirm a dose-dependent manner.

In figure 3C, no protein or mRNA level of mut16xG protein changed upon IGF2BP3 knockdown, which is not direct evidence to show mut16xG does not bind to IGF2BP2. RNA pull down or RIP-qPCR need to be performed.

Figure 4 shows IGF2BP paralogs regulate the protein level of FMRpolyG. What is the rationale to look for other IGF2BP paralogs??? Does mass spectrometry data also show the binding of FMRpolyG with IGF2BP family? Or, if no direct binding? What is the mechanism?

In figure5, all the inhibitors that the authors used are not directly targeting IGF2BP3, which made this part meaningless.

Reviewer #4

(Remarks to the Author)

Version 1:

Reviewer comments:

Reviewer #1

(Remarks to the Author)

The authors adequately addressed my comments, notably with addition of state of the art iPSC cell models. This work is now suitable for publication.

Reviewer #2

(Remarks to the Author)

In this revised manuscript, the authors made efforts to address several concerns raised in the previous review, including the addition of patient-derived cell models and a *C. elegans* model. While these new data and clarifications strengthen some aspects of the study, I remain concerned about the following points:

1. There is still no mechanistic evidence demonstrating how IGF2BP3 influences CGG repeat RNA activity or RAN translation. The authors argue that this is beyond the scope of the manuscript; however, given the central claim, mechanistic insight (e.g., effects on translation initiation, ribosomal recruitment, or RNA structure) is essential. The current finding that IGF2BP3 KD increases FMR1 mRNA is insufficient to explain functional consequences at the protein or phenotypic level.
2. While the inclusion of FXTAS patient iPSC-derived neurons is an improvement, the manuscript does not specify how many independent fragile X premutation iPSC lines were used. Current publication standards typically require at least three independent lines to ensure reproducibility. Furthermore, the reported necrosis phenotype in FXTAS iPSC-derived neurons has not, to my knowledge, been described previously, and there is insufficient discussion or validation of this novel observation.

3. The newly introduced *C. elegans* model expressing human FMR1 5'UTR with CGG repeats is intriguing; however, its characterization is insufficient. The authors do not clarify whether the observed toxicity is mediated by the RNA itself or by the translated polyG protein. A more detailed phenotypic and molecular characterization is needed to establish the relevance of this model to FXTAS.

4. The significantly enriched proteins in Supplemental Data 1 need to be listed.

Reviewer #3

(Remarks to the Author)

I can see authors has tried their best to answer my questions and concerns. I have no further concerns.

Reviewer #4

(Remarks to the Author)

Version 2:

Reviewer comments:

Reviewer #2

(Remarks to the Author)

The authors have addressed all my comments and this manuscript is ready for publication.

Reviewer #4

(Remarks to the Author)

Response do the Reviewer Comments

We thank Reviewers for the critical comments and experimental suggestions, which were instrumental in enhancing the quality of our research. We considered most of the advices in the revised version of the manuscript, which allowed us to significantly improve our work and further support our statement that IGF2BP proteins regulate the noncanonical translation of toxic polyglycine protein (FMRpolyG) from mutant FMR1 mRNA.

We have addressed all specific concerns below and made changes to the new version of the manuscript (indicated in red).

Reviewers' comments:

Reviewer #1 (Remarks to the Author):

Fragile X Tremor and Ataxia Syndrome (FXTAS) is a rare neurodegenerative disease caused by an expansion of CGG repeats located within the 5'UTR of the FMR1 gene. These CGG repeats are translated into a novel protein, FMRpolyG, which is prone to form cellular inclusions, and which expression is toxic in cell and animal models. Importantly, translation of these repeats depends on initiation at near-cognate start codons (GUG, ACG) located upstream of the repeats. Fidelity of translation initiation at near cognate codons vs canonical AUG start sites is closely regulated by several proteins (eIF1, eIF 2 and eIF5, among others), and thus of potential clinical interest in FXTAS to modulate expression of the toxic FMRpolyG protein.

In that aspect, Baud and collaborators found that the IGF2BP RNA binding proteins binds to the FMR1 5'UTR sequence and regulate FMRpolyG expression. This is an important finding as it open routes toward identifying a therapeutic approach for this devastating syndrome. Overall, presented data are solid and clear, experiments are technically well controlled, and the results are novel and of general interest for people working on FXTAS and/or on regulation of translation initiation. However, interest for this work is tempered by several points:

Main comments:

1. All data have been generated in vitro or using transformed cell lines (HeLa, HEK or SH-SY-5Y), thus this work, especially the use of drugs inhibiting IGF2BP as a potential therapeutic approach, suffers from a lack of validation in physiological iPS FXTAS cell or animal models.

We thank Reviewer for this crucial remark. In the current version of the manuscript, we have validated our findings in FXTAS patient-derived cells (fibroblasts, iPSC-derived neuronal progenitor cells and iPSC-derived neurons)- a new paragraph ***IGF2BP3 regulates the level of FMR1 mRNA in FXTAS patient derived cells; Figure 6***. In these models, we observed that silencing of IGF2BP3 resulted in increased relative *FMR1* mRNA levels, similarly as in cell lines with stable or transient expression of FMR99xG, what we showed in previous version of the manuscript. Importantly, we observed that IGF2BP3 KD rescued the necrosis phenotype in FXTAS iPSC-derived neurons. Additionally, we tested whether silencing of IGF2BP3 will decrease the number of FMRpolyG inclusions in FXTAS iPSC-derived neurons, and observed small but non-significant decrease (Fig. 6D). This can be explained by the fact that in this experimental set-up, siRNA targeting IGF2BP3 was added in cyclic manner, every 7 days, starting from day 7 after beginning of differentiation, thus the low level of IGF2BP3 was not constant during the experiment and in average was decreased by around 30% when comparing to control (Fig. S6D). On the other hand, IGF2BP3 level was decreased by around 70% comparing to control (Fig. S6B) in iPSC-driven neurons which were harvested 60h post siRNA treatment,. We have also observed that

BET inhibitors decreased the toxicity of CGGexp in FXTAS iPSC-derived neurons, but not in controls (Fig. S6G).

Additionally, we have generated a novel FXPAC animal model- *C. elegans* expressing RNA of human FMR1 5'UTR with 99 CGG repeats, fused to GFP at the 3' end, which exhibited phenotypic defects, described in a new paragraph ***imph-1/IGF2BP disruption rescues 99xCGG phenotype in Caenorhabditis elegans; Figure 7***. We observed that these phenotypes were rescued, when we crossed the 99xCGG strain with strain *imph-1Δ*, characterized by disrupted expression of IGF2BP ortholog. What is more, in the 99xCGG; *imph-1Δ* genetic background, we observed an increase in FMR1-99CGG RNA levels compared to the 99xCGG strain alone (Fig. 7C), consistent with the upregulation seen in FXTAS fibroblast and iPSC-derived neurons (Fig. 6A & 6B).

2. IGF2BP is a family of 3 proteins, hence effects of overexpressing or siRNA-mediated depletion of IGF2BP3 in Figure 2 could be fused to the analysis of IGF2BP1 & 2 in Figure 4. Importantly, siRNA-depletion needs to consider compensation between IGF2BP proteins, and a depletion of 2 or all 3 members by siRNA should be investigated. This is especially important as according to database, RNA expression of these proteins change according to cell type: only IGF2BP3 mRNA is expressed in SH-SY-5Y, but both IGF2BP1 and 3 transcripts are expressed in HeLa and all three are predicted to be expressed in HEK293, etc.). Thus, levels of these 3 proteins in the cell lines analyzed by the authors (with and without their corresponding siRNA treatment) should be presented. Moreover, out of the 3 IGF2BP proteins, only IGF2BP2 mRNA is predicted to be expressed in human brain, and only weakly (~10 nPTM in human cerebellum), thus this could be discussed for a future potential therapeutic strategy. Alternatively, as mRNA and protein levels not always correlate, a western on various human or mouse tissue, including the ones affected in FXTAS, showing expression of these protein would strengthen a proposed strategy based on decreasing IGF2BP proteins levels.

We thank Reviewer for this comment. In our manuscript we focus mostly on the role of IGF2BP3 paralog, therefore we decided to keep Figures 2 and 4 as separate parts, which, in our opinion, facilitates understanding of our work.

In the current version of the manuscript we investigated whether depletion of IGF2BP3 could be compensated by the increase in expression of another IGF2BP paralog and found, that only IGF2BP1 expression increases upon IGF2BP3 KD (Figure 4D). We also included the analysis of combined depletion of two or all three IGF2BP paralogs (Figure 4E), suggesting that IGF2BP2 and IGF2BP3 contribute most significantly to the regulation of FMRpolyG levels. Additionally, we now included the analysis of impact of BET inhibitors on the levels of IGF2BP1 and IGF2BP2 (Figure S5D). These changes, suggested by Reviewer, add a broader insight into the role of IGF2BP paralogs on regulation of FMRpolyG level.

3. Minor comments: the RNA structure presented in figure 3C would gain in clarity with symbols or colors indicating FMRpolyG and its near-cognate codons different from the color or symbol of other proteins/RAN product and their start sites. Same comment for the CA element required for IGF2BP binding, etc.

We have included changes suggested by the Reviewer in the current version of the manuscript.

4. Finally, various recent reports (including by the same Sobczak's group) show modulation of CGG repeat translation at the initiation, elongation or degradation level, notably by PSMB5 (Kong et al., 2022), RPS25/26 (Tutak et al., 2024), ANKZF1 (Tseng et al., 2024), etc. Thus, it could be of interest to show extend of the IGF2BP effect compared to other reported modulators.

As suggested, in the current version of the manuscript, we discuss different mechanisms of RAN translation modulation more extensively.

Reviewer #2 (Remarks to the Author):

The manuscript by Sobczak and colleagues investigates the role of insulin-like growth factor 2 mRNA-binding protein 3 (IGF2BP3) in the regulation of noncanonical translation of toxic proteins associated with CGG repeat expansions in the FMR1 gene, a mechanism implicated in fragile X-associated tremor/ataxia syndrome (FXTAS). The study demonstrates that IGF2BP3 binds to the 5' untranslated region (UTR) of FMR1 mRNA, specifically near expanded CGG repeats, and enhances repeat-associated non-AUG (RAN) translation, which produces neurotoxic polyglycine proteins (FMRpolyG). Below are some of my specific comments:

1. **In Vivo Validation of IGF2BP3 Binding to CGG Repeats:** The authors present comprehensive in vitro data showing IGF2BP3 binding to CGG repeats; however, without in vivo confirmation of this interaction, the physiological relevance remains uncertain. Demonstrating IGF2BP3 binding to CGG repeats within the context of living cells or in a relevant animal model would significantly strengthen their claims. Specifically, assays such as RNA immunoprecipitation (RIP) coupled with qPCR or cross-linking and immunoprecipitation (CLIP) would validate IGF2BP3's interaction with endogenous FMR1 mRNA. This data is essential to rule out potential artifacts from in vitro binding conditions, which may not fully mimic the cellular environment. The absence of in vivo validation leaves a critical gap, questioning the translatability of these findings to biological settings.

We thank Reviewer for this valuable suggestion. In the current version of the manuscript, we included eCLIP analysis of IGF2BP3 binding to 5'UTR of *FMR1* transcript in HepG2 cells, deposited in the ENCODE eCLIP resource (Nostrand et al., Nature Methods 2016). In this data, IGF2BP3 enrichment can be observed at the *FMR1* 5'UTR (Figure S1G), which validates our *in vitro* data.

2. **Limitations of Overexpression Systems in CGG Translation Assays:** The reliance on overexpression systems to study CGG repeat translation limits the applicability of the findings. Overexpression can alter cellular dynamics, potentially leading to non-physiological binding events or artifacts that would not occur at endogenous levels. Thus, it would be critical for the authors to assess RAN translation using native FMR1 transcript levels. Including in vivo or endogenously expressed FMR1 models would validate the observed regulatory effects and reinforce the physiological relevance of IGF2BP3's role in CGG translation. Without this, it is challenging to determine if the reported effects are an artifact of overexpression.

We thank Reviewer for this crucial remark. In the current version of the manuscript, we have validated our findings in FXTAS patient-derived cells (fibroblasts, iPSC-derived neuronal progenitor cells and iPSC-derived neurons)- a new paragraph ***IGF2BP3 regulates the level of FMR1 mRNA in FXTAS patient derived cells; Figure 6***. In these models, we observed that silencing of IGF2BP3 resulted in increased relative *FMR1* mRNA levels, similarly as in cell lines with stable or transient expression of FMR99xG, what we showed in previous version of the manuscript. Importantly, we observed that IGF2BP3 KD rescued the necrosis phenotype in FXTAS iPSC-derived neurons. Additionally, we tested whether silencing of IGF2BP3 will decrease the number of FMRpolyG inclusions in FXTAS iPSC-derived neurons, and observed small but non-significant decrease (Fig. 6D). This can be explained by the fact that in this experimental set-up, siRNA targeting IGF2BP3 was added in cyclic manner, every 7 days, starting from day 7 after beginning of differentiation, thus the low level of IGF2BP3 was not constant during the experiment and in average was decreased by around 30% when comparing to control (Fig. S6D). On the other hand, IGF2BP3 level was decreased by around 70% comparing to control (Fig. S6B) in iPSC-driven neurons which were harvested 60h post siRNA treatment,. We have also observed that BET inhibitors decreased the toxicity of CGGexp in FXTAS iPSC-derived neurons, but not in controls (Fig. S6G).

Additionally, we have generated a novel FXPAC animal model- *C. elegans* expressing RNA of human FMR1 5'UTR with 99 CGG repeats, fused to GFP at the 3' end, which exhibited phenotypic defects, described in a new paragraph ***imp-1/IGF2BP disruption rescues 99xCGG phenotype in Caenorhabditis elegans; Figure 7***. We observed that these phenotypes were rescued, when we

crossed the 99xCGG strain with strain *imph-1Δ*, characterized by disrupted expression of IGF2BP ortholog. What is more, in the 99xCGG; *imph-1Δ* genetic background, we observed an increase in FMR1-99CGG RNA levels compared to the 99xCGG strain alone (Fig. 7C), consistent with the upregulation seen in FXTAS fibroblast and iPSC-derived neurons (Fig. 6A & 6B).

3. Specificity of Small Molecule Inhibitors on IGF2BP Paralogs: While the authors indicate that small molecule inhibitors reduce IGF2BP3 expression, it remains unclear whether these inhibitors selectively target IGF2BP3 or also affect IGF2BP1 and IGF2BP2. Given the close structural similarity among IGF2BP family members, cross-reactivity is likely, and off-target effects could complicate the interpretation of results. Additional experiments are needed to dissect whether these inhibitors selectively target IGF2BP3 without interfering with other paralogs. For example, parallel knockdown or CRISPR knockout experiments of IGF2BP1 and IGF2BP2 would clarify if these paralogs contribute similarly to CGG repeat translation. This analysis needs to be more robust to the therapeutic implications of their findings, as off-target effects may reduce the efficacy or safety of IGF2BP3 inhibitors in clinical applications.

In the current version of the manuscript, we have included an analysis of the effect of small molecule inhibitors on the expression of all three IGF2BP paralogs (Figure S5D). As supposed by the Reviewer, mRNA level of all paralogs was decreased by administration of BET inhibitors. However, as shown now in Figure 4E, simultaneous KD of IGF2BP1, IGF2BP2 and IGF2BP3 has the greatest impact of FMR99xG expression, which may suggest that even if the level of all paralogs is decreased upon BET inhibitors administration, these small molecules may still have favourable effects in FXTAS cell models. What is more, our new data suggest the beneficial effect of JQ1 and IBET-151 on the apoptosis phenotype in FXTAS iPSC-derived neurons, strengthening our proposition that BET inhibitors could be new therapeutic option, in addition to mostly studied antisense oligonucleotide-based therapy. Nevertheless, off-target effects of BET inhibitors should be taken into account as written in the Discussion.

4. Mechanistic Insight into IGF2BP3's Role in RAN Translation: The manuscript presents IGF2BP3 as a regulator of RAN translation, yet it does not sufficiently explore the mechanistic basis underlying this regulation. Specifically, further investigation into whether IGF2BP3 impacts translation initiation factors, ribosomal recruitment, or RNA secondary structures would provide greater mechanistic clarity. This would also address whether IGF2BP3's influence is generalizable across other noncanonical translation events or specific to CGG repeats. Such insights are crucial for understanding the broader implications of IGF2BP3 function in RNA biology and neurodegenerative disorders.

We thank Reviewer for this comment. We believe that this investigation, although very interesting, is beyond the scope of the manuscript.

5. Experimental Controls and Replicability: There needs to be more discussion regarding experimental controls and biological replicates, particularly for complex assays like in vitro translation and protein-RNA interaction studies. Including appropriate controls—such as non-targeted RNA for pull-down assays and non-binding RNA mutants—would strengthen the reliability of the data. Additionally, clear reporting of the number of biological replicates would enhance reproducibility and lend robustness to the findings.

Conclusions included in the manuscript are based on at least three independent biological replicates, as stated below each figure. Non-targeted RNA (called "GC rich") control was included in pull-down assays (shown in Figure 1A-C), while non-targeted UAA control was included for in vitro binding assays: EMSA and FBA (shown in Figure S1F). In the current version of the manuscript, we have also included a novel proteomic data showing that IGF2BP3 was detected in pull-down of FMR1-99CGG RNA bait, but not synthetic 23CGG RNA bait (Figure S1D), which supports our finding that IGF2BP3 requires the appropriate flanking sequence, possibly with CA dinucleotide motif upstream of CGG repeats for binding (Figure 3D).

Reviewer #3 (Remarks to the Author):

The study explores the role of IGF2BP3, in aiding the translation of FMR1 mRNA with expanded CGG repeats. Using RNA-pull down followed by mass spectrometry and western blots, the authors identified IGF2BP3 binds to FMR1 mRNA probs either with or without CGG repeats. IGF2BP3 knockdown decreased the protein level of FMRpolyG but increased mRNA level, while OE of IGF2BP3 increased the protein level of FMR polyG but no changes on mRNA level. Then, by replacing the near-cognate ACG codon with AUG in FMRpolyG mRNA as a mutant FMR16xG, OE or KD of IGF2BP2 does not affect either protein or mRNA level of FMR16xG. Next, the authors showed that other IGF2BP family proteins including IGF2BP2 and IGF2BP1 also have similar effects in aiding FMRpolyG biosynthesis. Targeting IGF2BP3 by some BET inhibitors showed decreased protein levels of IGF2BP3 as well as FMR99xG. The whole part of the work stands overall on less solid ground, with unclear rationale for looking at sometimes FMR16xG but sometimes FMR99xG expression levels and with insufficient controls to determine that this is indeed the functional ability of IGF2BP3 to modulate FMRpolyG expression. In general, no experiment in the study is up to current technological standards, and no robust conclusion can be drawn in the absence of unbiased readouts, including at the very least measuring the levels of endogenous FMR1, FMR16xG, FMR96xG simultaneously upon KD or OE of IGF2BP2 in the key experiments, as well as the rescue experiments to confirm FMRpolyG is the essential target of IGF2BP2. Additional comments below, in no particular order of importance.

We thank Reviewer for this assessment. However, we hope that Reviewer will appreciate the fact that our conclusions are based on experiments using well established techniques, such as pull downs combined with high resolution mass spectrometry, biochemical assays (FBA, EMSA), gene silencing and overexpression in multiple cell models, including now also FXTAS patient-derived cells, that were analysed by western blots, flow cytometry, apoptosis/necrosis assays, confocal microscopy and quantitative real-time PCR. In our experimental design we paid attention to use orthogonal techniques in order to decrease the possibility of biased read-outs and artifacts.

Finally, as suggested by the Reviewer we performed simultaneous expression of FMR16xG and FMR99xG in cells with KD of IGF2BP3, but we were not able to draw any conclusions from this experiment since we observed overlap of the bands corresponding to both short and long polyglycine repeats (shown below).

1. In figure 1, the authors concluded that IGF2BP3 binds to the FMR1 5'UTR in a CGG-independent manner. To be convinced, biotinylated RNA probes designed with expanded CGG repeats but without FMR1 5' UTR region should be used as a negative control instead of GC-rich RNA in figure 1A, as well as for all the protein binding assay experiments in figure 1B-D.

As requested by the Reviewer, in the current version of the manuscript, we included a novel proteomic data showing that IGF2BP3 was detected in pull-down of FMR1-99CGG RNA bait, but not synthetic 23CGG RNA bait (Figure S1D), which supports our finding that IGF2BP3 binds to the FMR1 5'UTR independently of the CGG content.

Our statement that IGF2BP3 can bind to 5'UTR of FMR1 containing short CGG repeats is also supported by eCLIP analysis of IGF2BP3 binding to 5'UTR of *FMR1* transcript in HepG2 cells, containing normal size of CGG repeats, deposited in the ENCODE eCLIP resource (Nostrand et al., Nature Methods 2016). In this data, IGF2BP3 enrichment can be observed at the FMR1 5'UTR (Figure S1G).

2. Figure 1D, the authors turned to use FMR1-5'utr RNA containing 16CGG repeats in filter binding assay. Why not use FMR1-99CGG and FMR1-delCGG in this binding assay and do a comparison? What is the rationale to use FMR1-16CGG as research objective suddenly? In addition, only showing one Kd value means nothing. Please show the Kd value with either one negative or positive control.

Indeed, when we noticed that in pull-down assays IGF2BP3 binds to both FMR99xG and FMR1-delCGG (Figure 1C), and IGF2BP3 KD decreases levels of both FMR16xG and FMR99xG (Figure 2A), we decided to use FMR16xG RNA in *in vitro* assays. This was due to the fact, that gel purification and gel migration of transcripts containing 99 CGG repeats is very challenging, because of the stable secondary structure which hampers migration of the RNA in polyacrylamide. Similarly, mutagenesis and cloning of plasmids containing 99 CGG repeats is very difficult due to often occurring contractions of the repeats, therefore we performed mutagenesis on the FMR16xG plasmid. On the other hand, in IGF2BP3 KD assays, we tested the effect on both FMR16xG and FMR99xG to confirm *in vitro* data.

With respect to the Kd value of negative control (UAA) for EMSA and FBA assays (Fig. S1F), we were not able to determine the Kd since we did not observe any binding between IGF2BP3 and UAA RNA even in the highest, 100nM protein concentration. We can just say that $K_d \gg 100\text{nM}$.

3. Figure 2A, if as authors concluded that IGF2BP3 binds to FMRP in a CGG-independent manner, why is there no influence on normal FMRP protein level? how was the mRNA level of endogenous FMRP? The authors need to provide additional data to clarify this point.

We thank Reviewer for this important remark. Based on our data presented in the manuscript, we indeed concluded that IGF2BP3 binds to FMR1 5'UTR independently on the CGG repeats (Figure 1, Figure S1). This finding is also supported by IGF2BP3 eCLIP reads coverage of 5'UTR of FMR1 transcript (Figure S1G).

As requested by the Reviewer, we have measured the effect of IGF2BP3 KD on the mRNA level of endogenous *FMR1* in cell models with stable expression of FMR99xG (Figure 2A), control and FXTAS patients-derived fibroblasts (Figure 6A) as well as control and FXTAS iPSC-derived neurons (Figure 6B). In all these models, we observed an increase of relative *FMR1* mRNA level upon IGF2BP3 silencing, when comparing to control, suggesting that FMR1 is a target of IGF2BP3. However, as Reviewer noticed, IGF2BP3 KD impacted level of RAN-translated FMRpolyG but not FMRP. One possible explanation can be the fact that FMRP is translated from canonical AUG codon, and the translation efficiency is much higher than the efficiency of RAN translation of FMRpolyG. Therefore, binding of IGF2BP3 to FMR1 5'UTR will not influence canonical translation initiation of FMRP from optimal Kozak context. This explanation was added to the current version of the manuscript.

4. Figure 2D-F, the study should also perform with FMR16xG. Additionally, the apoptosis induced by FMR99xG upon KD of IGF2BP3 is not convinced unless rescue experiments are performed.

We thank Reviewer for the comment. However, we were not able to repeat experiments shown in Figures 2D-F for FMR16xG due to the fact that protein containing short polyglycine stretch (FMR16xG) does not form aggregates, in contrast to FMR99xG. We confirmed this fact in control iPSC-derived neurons, harboring 29CGG repeats in the 5'UTR of *FMR1* gene (Figure S6G). On the contrary, we observed FMRpolyG aggregates in FXTAS iPSC-derived neurons (Figure 6D). Additionally, in control iPSC-derived neurons, necrosis was not decreased upon IGF2BP3 KD (Figure S6F), suggesting that lack of FMRpolyG aggregates does not incur necrosis, as opposed to FXTAS iPSC-derived neurons, where FMRpolyG aggregates were detected.

5. RIP-seq should be performed to find the exact binding sites in FMRpolyG mRNA. Otherwise, can authors check the RIP-seq public data to find the motif?

IGF2BP3 is an RNA binding protein with multiple RNA targets. Schneider et al. (Nature communications, 2019) found that IGF2BP3 protein recognizes RNA containing CA- motifs and GGC core elements (either GGCA or CGGC) separated by appropriate spacing. In the 5'UTR of *FMR1* we identified three possible CA- followed by CGGC- motifs (shown in Fig. 3C). In line with that, we analysed the publicly available eCLIP data (Figure S1G) and found significant peaks of IGF2BP3 binding to the 5'UTR of *FMR1* mRNA, notably in the region where CA- and CGGC motifs occurs.

6. AUG-FMR16xG, normal FMRP should also be tested in figure 3A as controls.

We show the effect of IGF2BP3 KD on FMRP level in four different biological models: 1) HEK293 cells with stable expression of 16CGG repeats, 2) HEK293 cells with stable expression of 99CGG repeats- (shown Figure 2A), 3) HEK 293 cells with transient expression of 16CGG repeats, and 4) HEK293 cells with transient expression of 99CGG repeats (shown in Figure S2A). In neither case, we observed changes of endogenous FMRP level. These results, together with lack of effect of IGF2BP3 KD on FMRpolyG level produced from canonical AUG codon (Figure 3A), suggest that IGF2BP3 KD will possibly not influence level of FMR16xG produced from AUG codon based on canonical model of translation initiation.

7. In fig 3B, more IGF2BP3 concentrations should be added to confirm a dose-dependent manner.

We apologise the Reviewer for not fulfilling this request, which we now included in the discussion as a limitation of our study. Given the limited time, we decided to focus on validation of our initial findings, obtained *in vitro* and in overexpression systems, in FXTAS patient-derived cells and in novel FXTAS *C. elegans* model.

8. In figure 3C, no protein or mRNA level of mut16xG protein changed upon IGF2BP3 knockdown, which is not direct evidence to show mut16xG does not bind to IGF2BP2. RNA pull down or RIP-qPCR need to be performed.

We agree with the Reviewer that lack of this validation is a limitation of our study, which we now included in the discussion. However, we believe that including eCLIP data (Figure S1G) in the current version of the manuscript strengthens our conclusion that IGF2BP3 binds to 5'UTR of *FMR1*, also in an endogenous context.

9. Figure 4 shows IGF2BP paralogs regulate the protein level of FMRpolyG. What is the rationale to look for other IGF2BP paralogs??? Does mass spectrometry data also show the binding of FMRpolyG with IGF2BP family? Or, if no direct binding? What is the mechanism?

IGF2BP paralogs share overall high sequence identity- in the current version of the manuscript we included the alignment of IGF2BP1, IGF2BP2 and IGF2BP3 (Figure S4E) to better show similarities between paralogs. Additionally, IGF2BP paralogs may bind the same RNA target, i.e. CD44, MYC (reviewed in Ramesh-Kumar et al., Seminar in Cancer Biology, 2022). What is more, IGF2BP1 was also identified by us (Tutak et al., eLife 2025) and others (Malik et al., *EMBO Mol. Med*, 2021; Rosario et al., *FASEB J.*, 2022) in the pull-down screen of proteins binding to mutant *FMR1* 5'UTR RNA. Together, this prompted us to verify whether all IGF2BP paralogs can regulate FMRpolyG level.

10. In figure5, all the inhibitors that the authors used are not directly targeting IGF2BP3, which made this part meaningless.

We thank Reviewer for this comment. We would like to stress that in our manuscript we do not claim that the effect of BET inhibitors on IGF2BP3 is direct. Rather, we show our results as new interesting treatment possibility, which of course requires further investigation. Our point of view can be supported by the fact that BET inhibitors were used by different research teams to inhibit IGF2BP3 expression in various type of cancers (Mancarella et al., *Clin. Cancer Res.*, 2018; Coleman et al. *Sci. Rep.*, 2019; Dai

et al., *Nat. Commun.* 2024). It was shown that JQ1 impairs binding of bromodomain-containing protein 4 (BRD4) to chromatin in cancer cells, and CHIP-seq analysis revealed binding of BRD4 to IGF2BP3 promoter region in PC3 cells, which could be possible explanation of the negative effect of JQ1 on IGF2BP3 protein level.

Reviewer #4 (Remarks to the Author):

We thank Reviewer for supporting Early Career Researchers by training in the revision process.

Response to the Reviewer Comments

We thank Reviewer for the critical comments and experimental suggestions, which were instrumental in enhancing the quality of our research. We considered most of the advices in the revised version of the manuscript, which allowed us to significantly improve our work and further support our statement that IGF2BP proteins regulate the noncanonical translation of toxic polyglycine protein (FMRpolyG) from mutant FMR1 mRNA.

We have addressed all specific concerns below and made changes to the new version of the manuscript (indicated in blue).

Reviewer #2 (Remarks to the Author):

In this revised manuscript, the authors made efforts to address several concerns raised in the previous review, including the addition of patient-derived cell models and a *C. elegans* model. While these new data and clarifications strengthen some aspects of the study, I remain concerned about the following points:

1. There is still no mechanistic evidence demonstrating how IGF2BP3 influences CGG repeat RNA activity or RAN translation. The authors argue that this is beyond the scope of the manuscript; however, given the central claim, mechanistic insight (e.g., effects on translation initiation, ribosomal recruitment, or RNA structure) is essential. The current finding that IGF2BP3 KD increases FMR1 mRNA is insufficient to explain functional consequences at the protein or phenotypic level.

We thank Reviewer for this comment. In agreement with Senior Editor handling the manuscript, we did not seek further explanations of the mechanism of IGF2BP3 action on CGG repeat RNA activity or RAN translation. However, we completed the Discussion section with more explicit description of the limitations of our studies with regard to the mechanism by which IGF2BP3 regulated the noncanonical translation of the mutant FMR1 mRNA. We hope that we will be able to address these questions in our further research.

2. While the inclusion of FXTAS patient iPSC-derived neurons is an improvement, the manuscript does not specify how many independent fragile X premutation iPSC lines were used. Current publication standards typically require at least three independent lines to ensure reproducibility. Furthermore, the reported necrosis phenotype in FXTAS iPSC-derived neurons has not, to my knowledge, been described previously, and there is insufficient discussion or validation of this novel observation.

We thank Reviewer for this comment. Indeed, latest publications present at least two or three independent iPSC lines. In our work, we performed experiments on two independent FXTAS cell lines, expressing 81 CGG (FXTAS #1) or 72 CGG (FXTAS #2) repeats (Fig. 6B, S6B, S6D and Fig. 6D, S6H), while experiments shown in Fig. 6C and S6I were performed on one FXTAS cell line (72CGG). In the current version of the manuscript, we clarified these details in the Materials & Methods section, paragraph *IGF2BP3 regulates the level of FMR1 mRNA in FXTAS patient derived cells* and Figure 6 legend. However, we would like to point out that results presented in Fig.6 are

confirmatory to results shown in Figures 2 and 5, further supporting our claims about the role of IGF2BP3 in regulation of CGG RAN translation.

We have also referenced and discussed three publications describing the cell death (Sellier et al., 2017), reduced cell viability (Hoem et al., 2019) and neurotoxicity (Wright et al., 2022) caused by products of CGG translation. We believe that there is strong link between these reports and the necrosis phenotype in FXTAS iPSC neurons, which we observed.

To further support our claim, we have performed additional necrosis measurements in FXTAS iPSC-derived neurons, using as a positive control the antisense oligonucleotide (ASO) targeting directly CGGexp (ASO-CCG), which was previously published by us (Derbis et al., Nature Communications 2021). This experiment aligns with our findings described in Figure 6, and shows that ASO targeting expanded CGG repeats decrease necrosis in FXTAS iPSC-derived neurons, but not in control neurons. These experiments are an integral part of the findings of other research, which we submitted to Nucleic Acids Research. We are planning to resubmit in a few weeks.

Figure 1. Administration of antisense oligonucleotide targeting directly CGGexp lowers necrosis rate in FXTAS (72 CGG) iPSC-derived neurons. Necrosis was measured as fluorescence signals (relative fluorescence units). The graph presents relative mean values from N = 5 biologically independent samples treated with ASO Control (blue) or ASO CCG (red), with SDs.

3. The newly introduced *C. elegans* model expressing human FMR1 5'UTR with CGG repeats is intriguing; however, its characterization is insufficient. The authors do not clarify whether the observed toxicity is mediated by the RNA itself or by the translated polyG protein. A more detailed phenotypic and molecular characterization is needed to establish the relevance of this model to FXTAS.

We thank Reviewer for this comment. At this stage we cannot rule out the possibility that the observed phenotype is due to RNA toxicity. However, the fact that the FMRpolyG mRNA level increases in 99xCGG; *imph-1Δ* genotype (Fig. 7D) and yet the phenotype of animals is rescued (Fig. 7 E-G), suggest that FMRpolyG is the major contributor to the observed toxicity. This is further supported by our novel data showing that the fluorescence signal of FMRpolyG-GFP decreases in the head region of 99xCGG; *imph-1Δ* animals, comparing to 99xCGG strain (new Fig. 7C).

As requested by the Reviewer, we have also performed additional characterization of our animal model: quantification of FMRpolyG-GFP fluorescence in the head region of 99xCGG and 99xCGG; *imph-1Δ* animals (new Fig. 7C) and longevity assay of 99xCGG and 99xCGG; *imph-1Δ* animals (new Fig. 7G). These results further support our statement that toxic effect of rCGGexp expression is significantly reduced upon IGF2BP3 insufficiency.

4. The significantly enriched proteins in Supplemental Data 1 need to be listed.

We thank Reviewer for this comment. Significantly enriched proteins were listed in the previous version of Supplemental Data 1, however to increase the understanding of Datasheets, we have changed the names of Spreadsheets, added colour code of spreadsheets, and marked significantly enriched proteins in bold and borders.